# Dynamic Siamese Expansion Framework for Improving Robustness in Online Continual Learning

**Fei Ye**[1], **Yulong Zhao**[1], **Qihe Liu**[1]*, **Junlin Chen**[1], **Adrian G. Bors**[2]
**Jingling Sun**[1], **Rongyao Hu**[1], **Shijie Zhou**[1]
[1]School of Information and Software Engineering,
University of Electronic Science and Technology of China
[2]Department of Computer Science, University of York
{ feiye@uestc.edu.cn, yulongzhao913@outlook.com, qiheliu@uestc.edu.cn,
adrian.bors@york.ac.uk, jlsun@uestc.edu.cn, ryhu@uestc.edu.cn,
sjzhou@uestc.edu.cn }

## Abstract

Continual learning requires the model to continually capture novel information without forgetting prior knowledge. Nonetheless, existing studies predominantly address catastrophic forgetting, often neglecting enhancements in model robustness. Consequently, these methodologies fall short in real-time applications, such as autonomous driving, where data samples frequently exhibit noise due to environmental and lighting variations, thereby impairing model efficacy and causing safety issues. In this paper, we address robustness in continual learning systems by introducing an innovative approach, the Dynamic Siamese Expansion Framework (DSEF) that employs a Siamese backbone architecture, comprising static and dynamic components, to facilitate the learning of both global and local representations over time. Specifically, the proposed framework dynamically generates a lightweight expert for each novel task, leveraging the Siamese backbone to enable rapid adaptation. A novel Robust Dynamic Representation Optimization (RDRO) approach is proposed to incrementally update the dynamic backbone by maintaining all previously acquired representations and prediction patterns of historical experts, thereby fostering new task learning without inducing detrimental knowledge transfer. Additionally, we propose a novel Robust Feature Fusion (RFF) approach to incrementally amalgamate robust representations from all historical experts into the expert construction process. A novel mutual information-based technique is employed to derive adaptive weights for feature fusion by assessing the knowledge relevance between historical experts and the new task, thus maximizing positive knowledge transfer effects. A comprehensive experimental evaluation, benchmarking our approach against established baselines, demonstrates that our method achieves state-of-the-art performance even under adversarial attacks. Code is released at https://github.com/seSysdl/DSEF.

## 1 Introduction

Continual/Lifelong Learning (CL) has emerged as a pivotal subject within the domain of deep learning, significantly contributing to the progression of artificial intelligence systems [22]. In contrast to conventional deep learning methodologies, continual learning introduces a unique training framework wherein the model is exposed to a constrained set of samples, with prior data samples rendered inaccessible. This learning paradigm encounters a critical challenge termed catastrophic forgetting

---

*corresponding author

39th Conference on Neural Information Processing Systems (NeurIPS 2025).

[27], which can substantially impede the model's efficacy. This deterioration in performance occurs when the model modifies its parameters to assimilate new tasks.

Recent investigations into the mitigation of catastrophic forgetting in continual learning have delineated several strategic approaches, which are predominantly categorized into three principal domains: dynamic expansion methodologies [6, 14], which augment the model's capacity through the dynamic incorporation of additional hidden nodes and layers; memory-based approaches [5, 2], which enhance model efficacy by utilizing a judiciously curated set of samples retained within a memory buffer; and regularization techniques [19, 25], which typically integrate an auxiliary regularization term into the primary objective function to safeguard critical network parameters from substantial alterations. Among these methodologies, memory-based approaches exhibit efficacy in mitigating network forgetting when confronted with a constrained number of tasks, yet they frequently exhibit suboptimal performance in more challenging learning scenarios in which the number of tasks grows over time. Conversely, dynamic expansion methodologies are favored for their scalability and adaptability, rendering them apt for a diverse array of continual learning applications.

The majority of existing continual learning research presupposes that data samples are derived from the original data distribution [37]. Nonetheless, in more pragmatic scenarios such as autonomous driving, data samples frequently exhibit noise due to dynamically fluctuating illuminations, weather conditions, and road surfaces. Such noise-laden data samples can impair model performance, potentially leading to car accidents. This paper aims to enhance model robustness in continual learning by investigating a novel learning paradigm termed Online Continual Adversarial Defense (OCAD), wherein new data samples are encountered only once, and the model is expected to perform proficiently on both clean and adversarial samples post-training. OCAD presents three challenges: the adaptability to novel tasks (plasticity), the retention of antecedently acquired knowledge (stability), and the capability to counter adversarial samples (robustness). These challenges are mutual interaction during the training process, leading to significant performance degeneration for models.

To enhance plasticity, this study introduces an innovative Dynamic Siamese Expansion Framework (DSEF) that orchestrates and refines a Siamese backbone architecture to capture the semantically rich information. As a result, the Siamese backbone can help create a lightweight expert to adapt to a new task. The proposed Siamese backbone architecture comprises a static backbone for capturing global representations across all tasks and a dynamic backbone for delivering local representations, both of which are implemented using a pre-trained Vision Transformer (ViT) [8] to facilitate rapid adaptation. Moreover, the static and dynamic backbones predominantly share parameters to augment communication capabilities and diminish model complexity. Additionally, a learnable strategy network is proposed to ascertain and generate adaptive weights that delineate the significance of the static and dynamic backbones, thereby achieving optimal generalization performance.

To ensure robust stability, this paper introduces an innovative Robust Dynamic Representation Optimization (RDRO) methodology, which incrementally refines the dynamic backbone while preserving the static backbone in a fixed state throughout the optimization process. Specifically, the RDRO methodology formulates the static backbone as an auxiliary model that guides the optimization trajectory of the dynamic backbone through two regularization loss terms. The first loss term assesses the divergence between predictions of historical experts constructed from previously and currently acquired dynamic backbones, which ensures that updating the dynamic backbone does not precipitate substantial alterations in the prediction patterns of each historical expert. The subsequent loss term minimizes statistical discrepancies in the representations generated by previously and currently learned dynamic backbones, thereby preserving previously acquired robust representations.

To enhance adversarial robustness, we propose to integrate adversarial loss terms into the proposed RDRO framework to learn robust representations, ensuring optimal performance on both clean and adversarial samples. In addition, an innovative Robust Feature Fusion (RFF) methodology is introduced to amalgamate all previously acquired robust representations from historical experts with the representation extracted by the current expert, thereby facilitating the learning of new tasks. To optimize the positive transfer knowledge effects, the RFF method evaluates the knowledge similarity between each historical expert and the new task using a mutual information criterion, employing these metrics as adaptive weights in the feature fusion process. This strategy effectively reutilizes unactivated parameters and representations to enhance new task learning, resulting in superior generalization performance. A comprehensive series of experiments conducted across diverse datasets empirically demonstrates that the proposed approach achieves state-of-the-art performance.

The principal contributions of this research are delineated as follows : (1) This paper addresses a novel and challenging OCAD by proposing a novel DSEF that manages a Siamese backbone structure to capture global and local representations, enhancing plasticity; (2) This paper proposes a novel RDRO approach to regulate the optimization behaviour of the dynamic backbone by selectively minimizing the prediction and representation shifts of each history expert, which can prevent forgetting and maintain previously learned robust abilities; (3) This paper proposes a novel RFF approach to integrating all previously learned robust representations to promote the new task learning. Specifically, the proposed RFF approach evaluates the knowledge similarity between each history expert and the new task via a mutual information criterion, which provides adaptive weights for the feature fusion process, leading to better positive knowledge transfer effects.

## 2 Related Work

**Adversarial Defense.** Adversarial robustness has become a central concern in machine learning security, leading the field from early-stage heuristic defenses such as input preprocessing, generative noise suppression, and ensemble-based stabilization [35, 17, 1], to more principled and theoretically grounded approaches. Although initial methods offered short-term protection, they often lacked generalizability under adaptive attack scenarios. In contrast, adversarial training, which incorporates perturbed samples during model optimization, has demonstrated strong effectiveness and remains one of the most widely adopted defense strategies [11, 16, 23]. Additional techniques, including defensive distillation and robust knowledge transfer, have further enhanced model resilience against subtle and targeted manipulations [10, 33, 40]. Within the domain of continual learning, combining robustness with plasticity presents unique challenges. Recent studies have begun to explore this intersection by interpreting adversarial perturbations as structured task-like shifts, rather than treating them as isolated threats [39]. Building on this perspective, some methods embed adversarial training into architectures that expand over time, while employing feature and output distillation to prevent forgetting and preserve robustness across evolving tasks. This integrated direction offers a promising foundation for developing more secure and adaptable lifelong learning systems.

**Dynamic Expansion Model.** Lifelong learning has increasingly leveraged dynamic architectural strategies, where models evolve over time by integrating new neurons, layers, or specialized modules to handle incoming tasks. This structural plasticity allows for continual adaptation while minimizing interference with previously acquired knowledge by isolating task-specific components [6, 15, 28, 30, 34, 38, 18, 32]. Although convolutional neural networks (CNN) have traditionally served as the foundation for such approaches, the growing adoption of Vision Transformers (ViTs) reflects a broader shift toward architectures with greater capacity for scalability and flexible representation learning [8, 9]. Modern methods often incorporate modular attention mechanisms and decoupled task heads within ViT-based frameworks to better support incremental learning without performance degradation on earlier tasks [9, 36, 26]. Additionally, recent developments explore hybrid models that jointly optimize visual transformers with large-scale multimodal language models, aiming to improve both task transfer and generalization in dynamic settings [29]. Despite these innovations, many existing solutions remain primarily focused on preventing forgetting, with limited attention paid to adversarial robustness and resilience to distributional changes. More information can be found in **Appendix-A** from Supplementary Material (SM).

## 3 Methodology

### 3.1 Problem Statement

In continual learning, it is presumed that a model has access solely to a limited set of training samples for each task, while previous tasks are not accessible. The main goal of the model is to acquire new information without losing previously learned knowledge. Additionally, instead of most existing studies, which focus on a simple continual learning scenario, this paper explores a more intricate and realistic continual learning scenario referred to as Online Continual Adversarial Defense (OCAD), which introduces adversarial attacks aimed at undermining the model's performance. Consistent with the class-incremental framework, a training dataset $\mathcal{C}^s = \{\mathbf{x}_j, \mathbf{y}_j\}_{j=1}^{n^s}$ in the OCAD setting is divided into $N$ subsets $\{\mathcal{C}_1^s, \cdots, \mathcal{C}_N^s\}$ according to the category information, where $n^s$ denotes the total number of training data samples and each subset $\mathcal{C}_i^s$ contains data samples from one or several adjacent classes. $\mathbf{x}_j \in \mathcal{X}$ represents the $j$-th data sample, and $\mathbf{y}_j \in \mathcal{Y}$ denotes the corresponding class label. $\mathcal{X}$ and $\mathcal{Y}$ signify the data and label spaces, respectively. Each subset $\mathcal{C}_i^s$ is treated as a

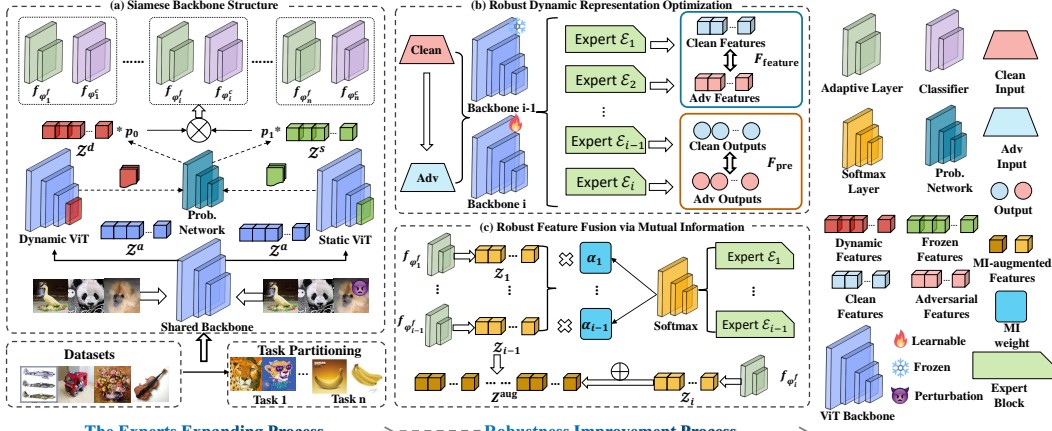

Figure 1: The comprehensive framework of the proposed DSEF including the RDRO and MBRFF mechanisms. The RDRO mechanism aligns the feature distributions and outputs of clean and adversarial samples in the siamese ViT network. Meanwhile, the MBRFF mechanism uses historical experts to augment feature extraction by mutual information.

specific task, denoted as $\mathcal{T}_i$. During the training process of a certain task ($\mathcal{T}_i$), the learning goal of a model is to find an optimal parameter set that minimizes the loss values of all previous and new data samples, expressed as :

$$\theta^\star = \underset{\theta \in \Theta}{\text{argmin}}\Big\{ \sum\nolimits_{c=1}^{j}\big\{\sum\nolimits_{t=1}^{|\mathcal{C}_c^s|}\{F_{ce}(\mathbf{y}_t, f_\theta(\mathbf{x}_t))\}\big\} \Big\}, \tag{1}$$

where $\Theta$ denotes the model's parameter space, and $|\mathcal{C}_c^s|$ represents the cardinality of the sample set $\mathcal{C}_c^s$. The cross-entropy loss is computed via the function $F_{ce}(\cdot)$. The intractability of identifying the optimal solution, as defined by Eq. (1), arises from the unavailability of data samples from all prior tasks. To mitigate this, existing studies have proposed the utilization of a fixed-size memory buffer [5] to preserve and replay critical past examples during the learning phase of a new task. When the new task learning is finished, the model's performance is evaluated across all testing datasets $\{\mathcal{C}_1^T, \cdots, \mathcal{C}_N^T\}$ using classification accuracy as the metric. In the OCAD framework's testing phase, the model's robustness is assessed via various adversarial attack methods, denoted as $\mathcal{A} = \{\mathcal{A}_1, \cdots, \mathcal{A}_T\}$. Each adversarial method $\mathcal{A}_j$ generates an adversarial dataset $\tilde{\mathcal{C}}_{i,j}^T = \mathcal{A}_j(\mathcal{C}_i^T, f_\theta)$ based on a testing dataset $\mathcal{C}_i^T$. The model's robustness is subsequently evaluated across all adversarial datasets $\{\tilde{\mathcal{C}}_{1,1}^T, \cdots, \tilde{\mathcal{C}}_{N,T}^T\}$ using classification accuracy metrics.

## 3.2   Siamese Backbone Structure

Existing studies in continual learning have investigated the efficacy of leveraging a pre-trained Vision Transformer (ViT) [8] backbone to enhance model performance. These methodologies typically employ a dynamic expansion framework, which utilizes a frozen ViT backbone to facilitate the construction of expert models. This design paradigm effectively preserves prior task knowledge by freezing all parameters of the pre-trained ViT but exhibits limitations in the context of novel task acquisition. To mitigate these constraints, this paper introduces a novel Siamese backbone architecture, which strategically employs both a static and a dynamic backbone to capture static and dynamic information, each instantiated with a pre-trained ViT. The static backbone maintains frozen parameters to furnish a generalized representation applicable across all tasks, whereas the dynamic backbone dynamically optimizes its parameters to adaptive representations. Given the substantial number of hidden layers and parameters inherent in the ViT-based backbone, updating all parameters would incur significant computational overhead. To address this, we propose training only the final $K$ representation layers of the dynamic backbone. Furthermore, the proposed Siamese backbone structure facilitates parameter sharing between the static and dynamic backbones, thereby minimizing redundant parameters and fostering inter-backbone communication.

Let $F_{\theta^a} : \mathcal{X} \to \mathcal{Z}^a$ represent a shared backbone, which processes a data sample $\mathbf{x}$ from the input space $\mathcal{X}^a$ and yields a feature representation $\mathbf{z}^a$ within the feature space $\mathcal{Z}^a$. Furthermore, let $F_{\theta^s} : \mathcal{Z}^a \to \mathcal{Z}$ and $F_{\theta^d} : \mathcal{Z}^d \to \mathcal{Z}$ denote the static and dynamic backbones, respectively, each of

which receives a feature vector extracted by $F_{\theta^a}$ and produces a representation $\mathbf{z}$ in the feature space $\mathcal{Z}$. By using the Siamese backbone architecture, an augmented representation can be formulated by :

$$\hat{\mathbf{z}} = F_{\theta^d}(F_{\theta^a}(\mathbf{x})) \otimes F_{\theta^s}(F_{\theta^a}(\mathbf{x})), \tag{2}$$

where $\otimes$ signifies the concatenation operation, which merges two feature representations. According to Eq. (2), the proposed methodology adaptively generates a novel expert to learn a new task, comprising a fully connected layer $F_{\varphi_j^f} : \mathcal{Z}^2 \rightarrow \mathcal{Z}'$ and a linear classifier $F_{\varphi_j^c} : \mathcal{Z}' \rightarrow \mathcal{Y}$, where $\varphi_j^f$ and $\varphi_j^c$ represent the parameters of the $j$-th expert. $\mathcal{Z}^2$ denotes the space of $\hat{\mathbf{z}}$ derived via the static and dynamic backbones. Furthermore, $\mathcal{Y}$ signifies the prediction space. The predictive function of the $j$-th expert is formulated as follows :

$$\mathcal{F}_{\mathrm{p}}(\mathbf{x}, \mathcal{E}_j) = F_{\varphi_j^c}\Big( F_{\varphi_j^f}\Big( F_{\theta^d}(F_{\theta^a}(\mathbf{x})) \otimes F_{\theta^s}(F_{\theta^a}(\mathbf{x}))\Big)\Big), \tag{3}$$

where $\{y_1', \cdots, y_M'\} = \mathcal{F}_{\mathrm{p}}(\mathbf{x}, \mathcal{E}_j)$ denotes the predicted probability vector and $M$ is the total number of classes.

**Learnable Strategy Network.** The representations delineated in Eq. (2) treat the static and dynamic backbones equivalently within the prediction process and thus would not yield optimal performance. Given that the static and dynamic backbones capture global and local representations, it is imperative to ascertain the significance of each representation autonomously, contingent upon the data's inherent characteristics. To this end, this study introduces a novel, learnable strategy network $F_{\gamma_j} : \mathcal{Z}^2 \rightarrow \mathcal{Y}'$ with the parameter set $\gamma_j$ for the $j$-th expert. This network processes a concatenated feature vector $\hat{\mathbf{z}}$, derived via Eq. (2), and subsequently outputs a selector probability vector over the space $\mathcal{Y}'$. Specifically, the predictive process of the $j$-th expert, facilitated by the learnable strategy network $F_{\gamma_j}$, is formalized as :

$$\mathcal{F}_{\mathrm{p}}'(\mathbf{x}, \mathcal{E}_j) = F_{\varphi_j^c}\Big( F_{\varphi_j^f}\Big( F_{\gamma_j}(\mathbf{x})[0]F_{\theta^d}(F_{\theta^a}(\mathbf{x})) \otimes F_{\gamma_j}(\mathbf{x})[1]F_{\theta^s}(F_{\theta^a}(\mathbf{x}))\Big)\Big), \tag{4}$$

where $F_{\gamma_j}(\mathbf{x})[0]$ and $F_{\gamma_j}(\mathbf{x})[1]$ denote the first and second dimensions of $F_{\gamma_j}(\mathbf{x})$. Compared to Eq. (3), the network $F_{\gamma_j}$ used in Eq. (4) can yield data-driven adaptive weights that determine the importance of static and dynamic backbones during the prediction process, which can achieve optimal performance.

### 3.3 Robust Dynamic Representation Optimization

Updating the parameters of the dynamic backbone $F_{\theta^d}$ is susceptible to detrimental knowledge transfer effects, given that all historical experts maintain parameter immutability throughout the learning phase of a novel task. To mitigate this, we introduce a novel methodology, termed Robust Dynamic Representation Optimization (RDRO), designed to optimize the dynamic backbone while minimizing catastrophic forgetting. Specifically, the objective of updating the dynamic backbone $F_{\theta^d}$ is to promote the new task learning and ensure the preservation of predictive capabilities and robust abilities acquired by each historical expert. To achieve this, the proposed RDRO approach minimizes the divergence between predictions generated using previously and currently acquired representations during the $j$-th task's learning phase, formally expressed as :

$$\begin{aligned}
F_{\mathrm{pre}} = \sum_{i=1}^{j-1} \Big\{ &F_{\mathrm{mse}}\Big( F_{\varphi_i^c}\Big( F_{\varphi_i^f}\Big( F_{\theta^d}(F_{\theta^a}(\mathbf{x})) \otimes F_{\theta^s}(F_{\theta^a}(\mathbf{x}))\Big)\Big), \\
&F_{\varphi_i^c}\Big( F_{\varphi_i^f}\Big( F_{\theta^s}(F_{\theta^a}(\mathbf{x})) \otimes F_{\theta^s}(F_{\theta^a}(\mathbf{x}))\Big)\Big)\Big)\Big\},
\end{aligned} \tag{5}$$

where $F_{\mathrm{mse}}(\cdot)$ signifies the Mean Squared Error (MSE) criterion. Nevertheless, the loss function delineated in Eq. (5) exclusively accounts for clean data samples, thereby disregarding adversarial data. Consequently, the dynamic backbone is rendered incapable of preserving the robust representation information. To mitigate this, the proposed RDRO methodology incorporates adversarial loss into Eq. (5), yielding :

$$\begin{aligned}
F_{\mathrm{pre}}' = \min_{\theta^d} \Big\{ F_{\mathrm{pre}}(\mathbf{x}) + \sum_{i=1}^{j-1} \Big\{ &F_{\mathrm{mse}}\Big( F_{\varphi_i^c}\Big( F_{\varphi_i^f}\Big( F_{\theta^d}(F_{\theta^a}(\mathbf{x})) \otimes F_{\theta^s}(F_{\theta^a}(\mathbf{x}))\Big)\Big), \\
&F_{\varphi_i^c}\Big( F_{\varphi_i^f}\Big( F_{\theta^s}(F_{\theta^a}(\mathbf{x})) \otimes F_{\theta^s}(F_{\theta^a}(\mathbf{x}))\Big)\Big)\Big) + \max_{||\mathbf{x}'-\mathbf{x}|| \leq \epsilon}\{F_{\mathrm{ce}}(\mathbf{y}, \mathcal{F}_{\mathrm{p}}'(\mathbf{x}', \mathcal{E}_j)\}\Big\}\Big\},
\end{aligned} \tag{6}$$

where $\mathbf{x}'$ signifies an adversarial instance of $\mathbf{x}$, synthesized via the expert $\mathcal{E}_j$, with $\epsilon$ representing the magnitude of the random vector perturbation. $F_{\text{ce}}$ is the cross-entropy loss function defined as :

$$F_{\text{ce}}(\mathbf{y}', \mathbf{y}) = \sum_{c=1}^{C'} \left\{ \mathbf{y}[c] \log(\mathbf{y}'[c]) \right\}, \tag{7}$$

where $\mathbf{y}[c]$ and $\mathbf{y}'[c]$ denote the $c$-th dimension of the class label $\mathbf{y}$ and the prediction $\mathcal{F}'_{\text{p}}(\mathbf{x}', \mathcal{E}_j)$, respectively. $C'$ represents the total number of categories. To mitigate catastrophic forgetting, the dynamic backbone's update must preserve feature statistical parity across experts during novel task acquisition. To achieve this, the proposed RDRO methodology initially generates two distinct feature vector sets for a given data batch $\mathbf{X} = \{\mathbf{x}_1, \cdots, \mathbf{x}_b\}$, leveraging the $i$-th historical expert, which is constructed using previously acquired and currently learned backbone parameters, expressed as :

$$\begin{aligned}
\mathbf{Z}^i &= \left\{ \mathbf{z}_t \mid \mathbf{z}_t = F_{\varphi_i^f}\Big( F_{\theta^d}(F_{\theta^a}(\mathbf{x}_t)) \otimes F_{\theta^s}(F_{\theta^a}(\mathbf{x}_t)) \Big), t = 1, \cdots, b \right\}, \\
\hat{\mathbf{Z}}^i &= \left\{ \mathbf{z}_t \mid \mathbf{z}_t = F_{\varphi_i^f}\Big( F_{\theta^s}(F_{\theta^a}(\mathbf{x}_t)) \otimes F_{\theta^s}(F_{\theta^a}(\mathbf{x}_t)) \Big), t = 1, \cdots, b \right\},
\end{aligned} \tag{8}$$

where $b$ denotes the batch size. In this study, we propose to formulate $\mathbf{Z}^i$ and $\hat{\mathbf{Z}}^i$ as distributions and minimize their probabilistic divergence as a regularization loss in the primary objective function. Specifically, we propose to employ Maximum Mean Discrepancy (MMD) [31] as the distance metric, owing to its facile implementation and the robust kernel-based theoretical foundation that facilitates formal analysis. The MMD criterion serves to quantify the discrepancy between two probability density functions. This distance measure is built on the embedding of probabilities within a Reproducing Kernel Hilbert Space (RKHS) [31]. Let $P(\mathbf{Z}^i)$ and $P(\hat{\mathbf{Z}}^i)$ denote Borel probability measures for $\mathbf{Z}^i$ and $\hat{\mathbf{Z}}^i$, respectively. We consider $\mathbf{z}^i$ and $\hat{\mathbf{z}}^i$ as random variables over a topological space $\mathcal{Z}^f$. We employ $\{f \in \mathcal{F} \mid f \colon \mathcal{X} \to \mathbf{R}\}$ to denote a function, with $\mathcal{F}$ representing a function class. The MMD criterion between $P(\mathbf{Z}^i)$ and $P(\hat{\mathbf{Z}}^i)$ is defined as [31].

$$\mathcal{L}_{\text{M}}(P(\mathbf{Z}^i), P(\hat{\mathbf{Z}}^i)) \triangleq \sup_{f \in \mathcal{F}} \left( \mathbb{E}_{\mathbf{z}^i \sim P(\mathbf{Z}^i)}\left[ f(\mathbf{z}^i) \right] - \mathbb{E}_{\hat{\mathbf{z}}^i \sim P(\hat{\mathbf{Z}}^i)}\left[ f(\hat{\mathbf{z}}^i) \right] \right). \tag{9}$$

where $\sup$ denotes the least upper bound of a set of numbers. If $P(\mathbf{Z}^i) = P(\hat{\mathbf{Z}}^i)$, we have $\mathcal{L}_{\text{M}}(P(\mathbf{Z}^i), P(\hat{\mathbf{Z}}^i)) = 0$. The function class $\mathcal{F}$ is considered as a unit ball in an RKHS with a positive definite kernel $k(\mathbf{x}, \mathbf{x}')$. Calculating Eq. (9) is usually computationally intractable. In practice, the MMD is estimated on the embedding space [21], expressed as :

$$\mathcal{L}_{\text{M}}^2(P(\mathbf{Z}^i), P(\hat{\mathbf{Z}}^i)) = \|\boldsymbol{\mu}_{P(\mathbf{Z}^i)} - \boldsymbol{\mu}_{P(\hat{\mathbf{Z}}^i)}\|^2, \tag{10}$$

where $\boldsymbol{\mu}_{P(\mathbf{Z}^i)}$ and $\boldsymbol{\mu}_{P(\hat{\mathbf{Z}}^i)}$ denote the mean embedding of $P(\mathbf{Z}^i)$ and $P(\hat{\mathbf{Z}}^i)$, respectively. $\| \cdot \|^2$ denotes the Euclidean distance. $\boldsymbol{\mu}_P$ is defined as $\boldsymbol{\mu}_{P(\mathbf{Z}^i)} = \int k(\mathbf{z}^i, \cdot) \frac{\partial P(\hat{\mathbf{Z}}^i)(\mathbf{z}^i)}{\partial \mathbf{z}^i} d\mathbf{z}^i$, where $P(\hat{\mathbf{Z}}^i)(\mathbf{z}^i)$ denotes the probability density function for $P(\mathbf{Z}^i)$. $\boldsymbol{\mu}_{P(\mathbf{Z}^i)}$ also satisfies $\mathbb{E}[f(\mathbf{z}^i)] = \langle f, \boldsymbol{\mu}_{P(\mathbf{Z}^i)} \rangle_{\mathcal{H}}$, where $\langle f, \cdot \rangle_{\mathcal{H}}$ denotes the inner product. Since RKHS has the reproducing property $f \in \mathcal{F}$, $f(\mathbf{z}^i) = \langle f, k(\mathbf{z}^i, \cdot) \rangle_{\mathcal{H}}$, Eq. (10) can be calculated using the kernel functions, expressed as :

$$\begin{aligned}
\mathcal{L}_{\text{M}}^2(P(\mathbf{Z}^i), P(\hat{\mathbf{Z}}^i)) = {} & \mathbb{E}_{\mathbf{z}^i, \mathbf{z}^{i'} \sim P(\mathbf{Z}^i)}[k(\mathbf{z}^i, \hat{\mathbf{z}}^i)] - 2\mathbb{E}_{\mathbf{z}^i \sim P(\mathbf{Z}^i), \hat{\mathbf{z}}^i \sim P(\hat{\mathbf{Z}}^i)}[k(\mathbf{z}^i, \hat{\mathbf{z}}^i)] \\
& + \mathbb{E}_{\hat{\mathbf{z}}^i, \hat{\mathbf{z}}^{i'} \sim P(\hat{\mathbf{Z}}^i)}[k(\hat{\mathbf{z}}^i, \hat{\mathbf{z}}^{i'})],
\end{aligned} \tag{11}$$

where $\mathbf{z}^{i'}$ and $\hat{\mathbf{z}}^{i'}$ are independent copies of $\mathbf{z}^i$ and $\hat{\mathbf{x}}^i$, respectively. In practice, we employ the same number of samples from $P(\mathbf{Z}^i)$ and $P(\hat{\mathbf{Z}}^i)$ ($N_{P(\mathbf{Z}^i)} = N_{P(\hat{\mathbf{Z}}^i)}$), where $N_{P(\mathbf{Z}^i)}$ and $N_{P(\hat{\mathbf{Z}}^i)}$ are the number of samples for $P(\mathbf{Z}^i)$ and $P(\hat{\mathbf{Z}}^i)$, respectively. Then Eq. (11) can be estimated using an unbiased empirical estimate, defined as :

$$\mathcal{L}_{\text{M}}^e(P(\mathbf{Z}^i), P(\hat{\mathbf{Z}}^i)) = \frac{1}{N_{P(\mathbf{Z}^i)}(N_{P(\mathbf{Z}^i)} - 1)} \sum_{i \neq j}^{N_{P(\mathbf{Z}^i)}} \left\{ h(i, j) \right\}, \tag{12}$$

where $h(i, j) = k(\mathbf{z}^i, \mathbf{z}^j) + k(\hat{\mathbf{z}}^i, \hat{\mathbf{z}}^j) - k(\mathbf{z}^i, \hat{\mathbf{z}}^j) - k(\mathbf{z}^j, \mathbf{z}^i)$. In addition, we also consider forming two groups of feature vectors using adversarial samples generated using the currently learned expert

at the $j$-th task learning, expressed as :

$$\mathbf{Z}^{i'} = \left\{ \mathbf{z}'_t \mid \mathbf{z}'_t = F_{\varphi_i^f}\left( F_{\theta^d}(F_{\theta^a}(\mathbf{x}_t)) \otimes F_{\theta^s}(F_{\theta^a}(\mathbf{x}'_t)) \right), \mathbf{x}'_t = \mathbf{x}_t + \bigtriangledown_{\mathbf{x}} F_{\text{ce}}(\mathcal{F}'_{\text{p}}(\mathbf{x}, \mathcal{E}_j), \mathbf{y}_t) \right\},$$
$$\hat{\mathbf{Z}}^{i'} = \left\{ \mathbf{z}'_t \mid \mathbf{z}'_t = F_{\varphi_i^f}\left( F_{\theta^s}(F_{\theta^a}(\mathbf{x}_t)) \otimes F_{\theta^s}(F_{\theta^a}(\mathbf{x}_t)) \right), \mathbf{x}'_t = \mathbf{x}_t + \bigtriangledown_{\mathbf{x}} F_{\text{ce}}(\mathcal{F}'_{\text{p}}(\mathbf{x}, \mathcal{E}_j), \mathbf{y}_t) \right\},$$
(13)

Let $P(\mathbf{Z}^{i'})$ and $P(\hat{\mathbf{Z}}^{i'})$ represent two Borel probability measures for $\mathbf{Z}^{i'}$ and $\hat{\mathbf{Z}}^{i'}$, respectively. Based on the MMD criterion, the proposed RDRO approach includes a regularization loss term for the representations at the $j$-th task learning, expressed as :

$$F_{\text{feature}} = \min_{\theta^d} \left\{ \frac{1}{j-1} \sum_{i=1}^{j-1} \left\{ \mathcal{L}_{\text{M}}^e(P(\mathbf{Z}^i), P(\hat{\mathbf{Z}}^i)) + \mathcal{L}_{\text{M}}^e(P(\mathbf{Z}^{i'}), P(\hat{\mathbf{Z}}^{i'})) \right\} \right\}.$$
(14)

Based on the loss terms defined in Eq. (6) and Eq. (14), the final objective function for optimizing the dynamic backbone is expressed as :

$$F_{\text{RDRO}} = F_{\text{feature}} + F'_{\text{pre}}.$$
(15)

Furthermore, given the static backbone's immutable nature, its utilization in regulating the dynamic backbone's optimization may engender over-regularization, thereby constricting the capacity for novel task acquisition. To mitigate this, the proposed RDRO methodology effectuates a weight transfer from the dynamic backbone to the static backbone subsequent to each task transition. This design facilitates the incremental preservation of novel information within the static backbone, consequently alleviating over-regularization phenomena.

## 3.4 Robust Feature Fusion via Mutual Information

Many existing studies in continual learning usually utilize all active parameters to facilitate new task learning, often disregarding previously acquired representations. The utilization of critical historical representations is posited to engender positive knowledge transfer effects, thereby enhancing performance. To this end, this paper introduces a novel Mutual Information-Based Robust Feature Fusion (MBRFF) approach, which automatically ascertains knowledge similarity between each historical expert and the new task via a mutual information criterion. Specifically, during a given task learning phase ($\mathcal{T}_j$), the proposed MBRFF approach initially establishes the joint distribution $P(\mathbf{Y}^i, \mathbf{Y})$, where $P(\mathbf{Y})$ and $P(\mathbf{Y}^i)$ represent the marginal distributions of the true class labels and the corresponding predictions made using the $i$-th expert, respectively. Let $\mathbf{Y}^i$ and $\mathbf{Y}$ denote the random variables of the joint distribution $P(\mathbf{Y}^i, \mathbf{Y})$. The mutual information between $\mathbf{Y}^i$ and $\mathbf{Y}$ is defined as follows :

$$I(\mathbf{Y}^i; \mathbf{Y}) = \sum_{\mathbf{y}^i \in \mathbf{Y}^i} \left\{ \sum_{\mathbf{y} \in \mathbf{Y}} \left\{ P(\mathbf{Y}^i, \mathbf{Y})(\mathbf{y}^i, \mathbf{y}) \log \frac{P(\mathbf{Y}^i, \mathbf{Y})(\mathbf{y}^i, \mathbf{y})}{p(\mathbf{Y}^i)(\mathbf{y}^i) p(\mathbf{Y})(\mathbf{y})} \right\} \right\},$$
(16)

where $P(\mathbf{Y}^i, \mathbf{Y})(\mathbf{y}^i, \mathbf{y})$ signifies the probability density function of $P(\mathbf{Y}^i, \mathbf{Y})$. The mutual information term $I(\mathbf{Y}^i; \mathbf{Y})$, as defined in Eq. (16), quantifies the degree of familiarity exhibited by the $i$-th expert concerning the novel task $\mathcal{T}_j$. To mitigate potential numerical overflow, the proposed MBRFF methodology normalizes the mutual information terms, subsequently employing them as adaptive weights to modulate the significance of each historical expert during the learning phase of a new task, as articulated by :

$$\alpha_i = \frac{\exp(I(\mathbf{Y}^i; \mathbf{Y}))}{\sum_{c=1}^{j-1} \{\exp(I(\mathbf{Y}^c; \mathbf{Y}))\}},$$
(17)

where $\exp(\cdot)$ is the exponential function and $\alpha_i$ is the adaptive weight for the $i$-th expert. By utilizing Eq. (17), we can integrate representations from all history experts to form an augmented representation, expressed as :

$$\mathbf{Z}^{\text{aug}} = \sum_{i=1}^{j-1} \left\{ \alpha_i F_{\varphi_j^f}\left( F_{\gamma_i}(\mathbf{x})[0] F_{\theta^d}(F_{\theta^a}(\mathbf{x})) \otimes F_{\gamma_i}(\mathbf{x})[1] F_{\theta^s}(F_{\theta^a}(\mathbf{x})) \right) \right\}.$$
(18)

Based on the augmented representations defined in Eq. (18), the prediction process of the $j$-th expert can be expressed as :

$$\mathcal{F}'_{\text{aug}}(\mathbf{x}, \mathcal{E}_j) = F_{\varphi_j^c}\left( \mathbf{Z}^{\text{aug}} \otimes F_{\varphi_j^f}\left( F_{\gamma_j}(\mathbf{x})[0] F_{\theta^d}(F_{\theta^a}(\mathbf{x})) \otimes F_{\gamma_j}(\mathbf{x})[1] F_{\theta^s}(F_{\theta^a}(\mathbf{x})) \right) \right).$$
(19)

Compared to Eq. (4), the prediction process defined in Eq. (19) involves all previously learned robust representations and thus can achieve robust predictions. The pseudocode can be found in **Appendix-B** from the Supplementary Material (SM).

| Split CIFAR-10 | | | | | | | |
|---|---|---|---|---|---|---|---|
| **Methods** | Refresh | Refresh (Adv) | DER | DER(Adv) | DER++ | DER++ (Adv) | AIR | DSEF |
| Clean | **92.47%** | 91.76% | 92.46% | 91.49% | 91.42% | 91.70% | 49.80% | 90.72% |
| FGSM | 55.84% | 58.09% | 55.51% | 60.78% | 56.13% | 42.39% | 18.64% | **82.36%** |
| PGD | 05.32% | 06.43% | 05.79% | 07.29% | 05.29% | 06.23% | 03.92% | **79.79%** |
| PGDL2 | 65.87% | 68.64% | 64.42% | 69.58% | 64.28% | 52.17% | 22.34% | **82.43%** |
| BIM | 48.69% | 47.96% | 50.60% | 48.79% | 47.63% | 48.47% | 16.65% | **87.43%** |
| CW | 00.39% | 00.34% | 00.39% | 00.19% | 00.27% | 00.76% | 00.29% | **82.43%** |
| AutoAttack | 03.17% | 04.79% | 02.06% | 02.77% | 02.14% | 03.87% | 00.76% | **90.90%** |
| Average | 38.82% | 39.71% | 40.12% | 41.77% | 38.16% | 35.08% | 16.05% | **85.15%** |

| Split CIFAR-100 | | | | | | | |
|---|---|---|---|---|---|---|---|
| **Methods** | Refresh | Refresh(Adv) | DER | DER(Adv) | DER++ | DER++ (Adv) | AIR | DSEF |
| Clean | 62.24% | 61.37% | 52.79% | 48.67% | 57.74% | 57.49% | 23.79% | **68.17%** |
| FGSM | 25.89% | 27.42% | 21.49% | 21.09% | 22.95% | 18.74% | 09.43% | **51.71%** |
| PGD | 03.29% | 04.94% | 03.76% | 05.27% | 04.16% | 04.32% | 01.46% | **44.79%** |
| PGDL2 | 32.96% | 34.47% | 27.68% | 25.74% | 30.73% | 24.49% | 12.93% | **53.42%** |
| BIM | 25.47% | 24.17% | 22.49% | 21.36% | 22.98% | 23.18% | 10.27% | **60.79%** |
| CW | 00.58% | 00.29% | 00.59% | 00.94% | 00.56% | 00.79% | 00.31% | **52.68%** |
| AutoAttack | 02.34% | 03.28% | 02.44% | 03.73% | 02.84% | 03.07% | 00.89% | **66.52%** |
| Average | 21.82% | 22.27% | 18.74% | 18.11% | 20.28% | 18.86% | 08.44% | **56.86%** |

Table 1: The classification accuracy of the standard datasets under clean and adversarial conditions.

# 4 Algorithm Implementation

The comprehensive learning pipeline of our proposed method is illustrated in Fig. 1. The overall procedure can be decomposed into four key steps:

**Step 1: Model expansion process** During the learning of the first task, we construct a shared backbone $F_{\theta^a}$, which serves as the foundation for expert module creation. In addition, we initialize a dynamic backbone $F_{\theta^d}$ and a static backbone $F_{\theta^s}$, which together constitute a Siamese network architecture. For each subsequent task $C_i$, a new expert module $\mathcal{E}_i$ is dynamically instantiated to accommodate task-specific knowledge.

**Step 2: Calculate robust optimization loss** We begin by obtaining data samples from the current task, which are first processed by the foundational backbone to extract initial representations. These representations are then forwarded through both the dynamic and static backbones, resulting in corresponding feature vectors and predictions. To enhance robustness, we compute the optimization loss terms using Eq. 6 and Eq. 14, which guide the learning of both prediction accuracy and feature consistency.

**Step 3: Mutual information fusion** To enhance the predictive capacity of the current expert, we additionally compute the outputs of historical experts and evaluate their relevance using mutual information. The importance weights derived from this process are used to guide the aggregation, as formalized in Eq. 19.

**Step 4: Optimizing the model's parameters.** The primary objective function for training the $i$-th expert at the $i$-th task learning, involves the RDRO loss terms, expressed as :

$$F(\mathbf{x}, \mathbf{y}, i) = \min_{\theta^a, \theta^s, \theta^d} \left\{ \sum_{c=1}^{b} \{F_{\mathrm{p}}(\mathbf{x}_c, \mathbf{y}_c) + F_{\mathrm{p}}(\mathbf{x}'_c, \mathbf{y}_c)\} + \lambda F_{\mathrm{RDRO}} \right\}, \tag{20}$$

where $F_{\mathrm{p}}(\mathbf{x}_c, \mathbf{y}_c)$ represents the evaluation function that compares the prediction obtained from Eq. 4 with the ground-truth label $\mathbf{y}$, and $\mathbf{x}'_c$ means the adversarial sample. $F_{\mathrm{RDRO}}$ is defined in Eq. 15, and $\lambda$ is the hyperparameter.

# 5 Experiment

## 5.1 Experimental Setting

**Baselines:** In this section, we present a thorough comparison between our proposed method and several established continual learning baselines, with a primary focus on experience replay-based

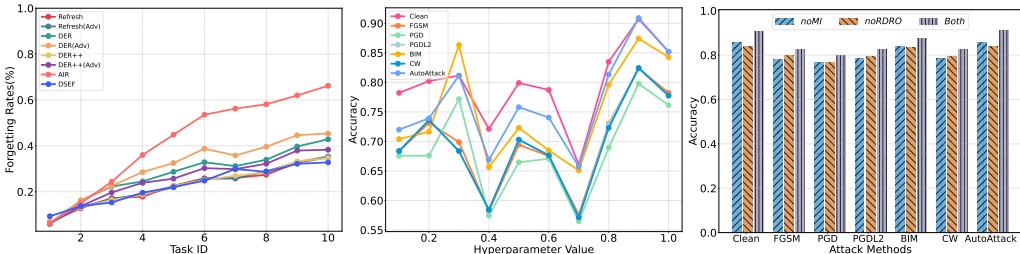

(a) The forgetting curve.    (b) The hyperparameter $\lambda$ analysis.    (c) Different configurations.

Figure 2: (a) The comparison of the forgetting curves between DSEF and other baseline methods after learning a sequence of tasks. (b) The model's performance when varying $\lambda$ from Eq. (20). (c) The performance of the proposed DSEF with different configurations.

| | | | | Split CUB200 | | | | |
|---|---|---|---|---|---|---|---|---|
| **Methods** | Refresh | Refresh (Adv) | DER | DER(Adv) | DER++ | DER++ (Adv) | AIR | DSEF |
| Clean | **67.62**% | 61.43% | 58.75% | 46.55% | 65.03% | 53.47% | 29.37% | 58.47% |
| FGSM | 26.84% | **28.52**% | 21.84% | 19.83% | 25.18% | 20.82% | 10.48% | 26.79% |
| PGD | 00.46% | 00.96% | 00.47% | 00.56% | 00.42% | 00.52% | 00.21% | **19.83**% |
| PGDL2 | 37.85% | **39.76**% | 32.17% | 27.37% | 35.28% | 28.94% | 15.58% | 27.48% |
| BIM | 22.17% | 18.74% | 18.16% | 15.83% | 21.74% | 17.12% | 08.33% | **34.85**% |
| CW | 05.16% | 03.56% | 04.32% | 05.72% | 04.76% | 04.25% | 04.15% | **26.23**% |
| AutoAttack | 00.21% | 00.15% | 00.21% | 00.29% | 00.17% | 01.65% | 09.74% | **40.74**% |
| Average | 22.90% | 21.87% | 19.41% | 16.59% | 21.79% | 18.11% | 11.12% | **33.48**% |
| | | | | Split TinyImageNet | | | | |
| **Methods** | Refresh | Refresh(Adv) | DER | DER(Adv) | DER++ | DER++ (Adv) | AIR | DSEF |
| Clean | 63.28% | 62.36% | 54.32% | 52.62% | **63.36%** | 60.26% | 30.27% | 60.21% |
| FGSM | 25.84% | 25.17% | 21.68% | 18.97% | 26.42% | 18.47% | 11.75% | **45.34**% |
| PGD | 02.38% | 03.12% | 01.97% | 02.65% | 02.46% | 02.13% | 00.82% | **40.13**% |
| PGDL2 | 33.78% | 34.58% | 31.34% | 28.18% | 33.48% | 25.17% | 13.47% | **47.78**% |
| BIM | 22.46% | 22.84% | 18.57% | 19.67% | 23.52% | 19.42% | 10.94% | **55.73**% |
| CW | 00.64% | 00.42% | 00.65% | 00.57% | 00.69% | 00.67% | 00.34% | **47.77**% |
| AutoAttack | 01.12% | 01.25% | 00.82% | 01.14% | 00.94% | 00.86% | 00.34% | **60.80**% |
| Average | 21.35% | 21.39% | 18.47% | 17.68% | 21.55% | 18.14% | 09.70% | **51.10**% |

Table 2: The classification accuracy of the complex datasets under clean and adversarial conditions.

approaches. The methods evaluated include Refresh [13], DER, and DER++ [3], all of which utilize a fixed backbone throughout the training process. Since our framework incorporates adversarial training, we additionally assess the adversarial variants of these baselines, namely Refresh (Adv), DER (Adv), and DER++ (Adv), to evaluate their performance under adversarial scenarios. We also include AIR [39], a recent method designed specifically for continual adversarial defense, which treats each new class as an independent task. For a fair comparison, all replay-based methods are configured with an identical memory buffer size of 500 samples. Further details on the experimental setup can be found in **Appendix-C** from the Supplementary Material (SM).

**Metrics:** To systematically compare the effectiveness of different continual learning methods under adversarial settings, we adopt classification accuracy as the core performance metric across a range of training environments. After gathering all experimental outcomes, we compute an overall average accuracy for each method by aggregating results across multiple attack scenarios. To provide a comprehensive assessment of model robustness, we consider a total of seven adversarial attack strategies. These include the Fast Gradient Sign Method (FGSM) [12], Projected Gradient Descent (PGD) [24], PGD with $L_2$ norm, the Basic Iterative Method (BIM) [20], the Carlini and Wagner attack (CW) [4], and AutoAttack [7], which integrates several strong attacks in an ensemble fashion.

### 5.2 Evaluation on Standard Datasets

Table 1 presents the classification results on Split CIFAR-10 and Split CIFAR-100, comparing our proposed method with a range of state-of-the-art continual learning techniques. The empirical evidence indicates that our approach consistently delivers superior performance across both

datasets, exhibiting stronger robustness against most adversarial attack methods. While the clean accuracy of our model may be marginally lower than that of certain baselines that do not incorporate adversarial defense, our method significantly outperforms those relying on adversarial training strategies. This suggests that our framework achieves a favorable trade-off between maintaining accuracy on clean data and enhancing robustness under adversarial conditions. Notably, our method attains the highest average accuracy when considering both clean and adversarial samples, further underscoring its effectiveness in balancing standard performance and security.

## 5.3 Evaluation on Complex Datasets

To further assess the generalizability and robustness of different methods, we conduct experiments on more challenging benchmarks, namely Split CUB200 and Split TinyImageNet. The results are summarized in Table 2. On Split CUB200, although our method performs slightly below certain baselines in a few specific cases, it ultimately achieves the highest overall performance in terms of the average accuracy metric. This highlights its ability to maintain stable performance across diverse conditions. For the Split TinyImageNet

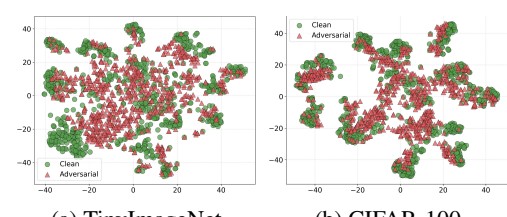

(a) TinyImageNet     (b) CIFAR-100

Figure 3: t-SNE visualization of clean vs. adversarial samples.

dataset, our approach consistently outperforms all competing methods across all evaluated settings, including the average score, demonstrating its strong adaptability and effectiveness in more complex continual learning scenarios.

## 5.4 Analysis Study

**t-SNE Visualization.** In our DSEF framework, the shared backbone is utilized to extract deep features from both clean and adversarial inputs. To better understand the impact of adversarial perturbations on feature representations, we employ t-SNE for dimensionality reduction and visualization. As illustrated in Fig. 3, the resulting embeddings show that clean and adversarial examples are closely clustered and largely overlapping in the feature space. This suggests that the backbone network is capable of mapping both types of samples into a consistent and robust representation space. Such behavior is driven by the proposed RDRO mechanism, which explicitly promotes representation invariance across different input domains. As a result, expert modules that operate on top of the shared backbone can establish more reliable decision boundaries, ultimately enhancing the model's robustness and classification accuracy. More ablation results can be found in **Appendix-D** from Supplementary Material (SM).

## 6 Conclusion and Limitation

This paper introduces a novel DSEF framework to enhance robustness in online continual learning by integrating a Siamese backbone with static and dynamic components. A Robust Dynamic Representation Optimization (RDRO) method is proposed to regulate dynamic updates while preserving prior knowledge. Additionally, a Mutual Information-Based Robust Feature Fusion (MBRFF) is proposed to adaptively reuse historical expert knowledge. Experiments on various benchmarks demonstrate that DSEF achieves superior performance under both clean and adversarial conditions, showcasing its effectiveness in addressing forgetting and robustness simultaneously. The primary limitation of this paper is that we adopt several popular adversarial attack methods in the experiment. In our future study, we will explore more recent adversarial attack methods to evaluate the model's performance.

## 7 Acknowledgments and Disclosure of Funding

This study is supported by grants from the National Natural Science Foundation of China (Grant No. 62506067, No. 62306066), the Fundamental Research Funds for the Central Universities (Grant No. ZYGX2025XJ024, No. ZYGX2025XJ025) and Sichuan Provincial Natural Science Foundation Project (No.2025ZNSFSC0510).

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
