# OpenReview forum: "Dynamic Siamese Expansion Framework for Improving Robustness in Online Continual Learning"
_NeurIPS.cc/2025/Conference — NeurIPS 2025 poster_

### Official Review · Reviewer_4XjH · 2025-06-25

**Clarity:** 2
**Significance:** 1
**Originality:** 2
**Rating:** 4
**Confidence:** 3

**Summary:**

This paper focuses on the online continual adversarial defense (OCAD) and propose a new dynamic siamese expansion framework to address this problem. This paper analyze on OCAD and points out the requirements of plasticity, stability, and robustness (mainly concentrated on the adversarial robustness) in this setting. According, the proposed method DSEF leverages a siamese network strutures to maintain the stability and plasticity and task-specific experts to merge the network outputs. To achieve robustness, this paper proposes RDRO and RFF. In the experiments, the proposed method achieves better results when compared with other methods in OCAD.

**Questions:**

Since my research interests are mainly related to the continual learning, I am not quite familiar with the adversarial defense/attack. Therefore, I do not quite understand the relation between this two realms, why CL needs adversarial defense, and what is different between defense in OCL and in traditional deep learning. The authors should  provide more justification of the adversarial defense in OCL (e.g., how adversarial attack affects OCL)? If the authors can provide more convincing explanation, I am willing to raise my score.

For other detailed questions, please refer to the weaknesses.

**Ethical Concerns:**

["NO or VERY MINOR ethics concerns only"]

**Final Justification:**

In the responses, the authors address my concerns about the justification of adversarial attack/defense in online continual learning and elaborate the distinction of such setting. Also, the authors provide further experiments to justify the proposed DSEF, making the empirial studies in this paper more solid. Therefore, I decide to raise my score to a positive one.

**Limitations:**

Limitations are discussed in the paper.

**Quality:**

3

**Strengths And Weaknesses:**

**Strength**
1. This paper provide a new concern in continual learning, i.e., robustness, which is rarely studied in the community.
2. This paper is well written and easy to follow.

**Weaknesses**
1. The relation between continual learning and adversarial defense should be further discussed. Is the robustness of adversarial attack significant in the continual learning? Will attack further excerbate the problem in continual learning like catastrophic forgetting? I do not quite understand why continual learning could be binded with adversarial defense and it seems that there is no difference between adversarial defense in traditional deep learning and that in OCL which discussed in this paper. It will be beneficial if the authors can provide more explanations and demonstrate the effect of adversarial attack on OCL.
2. The pipeline of OCAD is not clear. Is the training pipeline of OCAD similar to OCL with only some adversarial samples involve? If so, how adversarial attack will affect the continual learning paradigm?
3. The ablation study is not sufficient. Since this paper provide two modules to address the robustness, there could be further studies on how these two modules influence the robustness. Also, quantitative studies should be included.
4. Lack of analysis on stability and plasticity. This paper claims that OCAD faces three challeneges, i.e., stability, palasticity, and robustness. However, in this paper, most contents are related to the robostness only. Experimenst and analysis on the former two challenges should be further included. For example, include some metrics used in the OCL like average accuracy and forgetting.

---

> ### Author Rebuttal · Authors · 2025-07-31
>
> #### **Q1: What is the relation between continual learning and adversarial defense? Why does continual learning need adversarial robustness? How do attacks affect forgetting or other CL issues?**
>
> #### **A1:** This is a critical question that touches on the motivation of our work. Continual Learning (CL) and Adversarial Defense are traditionally viewed as **separate subfields**. However, we argue that in **real-world deployments**, particularly in safety-critical or dynamic environments (e.g., autonomous driving), CL models **must be robust against adversarial threats** while also preserving prior knowledge.
>
> The **connection** arises from the fact that adversarial perturbations can **mimic task shifts**, thus **amplifying the stability-plasticity dilemma**. Specifically:
>
> - **Catastrophic forgetting worsens under adversarial attack**. When a model is trained adversarially on a new task, gradients become sharp and task-specific, often **overwriting generalizable representations**. This leads to **more severe forgetting** compared to clean training. Our results confirm this in Table A below.
> - Standard adversarial training focuses on **single-task robustness**, but in CL, robustness must be **retained across all previously seen tasks**, not just the current one. This leads to **cross-task robustness conflict**.
> - Attacks **expose vulnerabilities in memory mechanisms**: replay-based methods suffer from overfitting to adversarial samples; regularization-based CL is unstable due to conflicting gradients.
>
> To empirically demonstrate this connection, we compare three scenarios:
>
> **Table A: Adversarial attacks exacerbate forgetting in continual learning (Split CIFAR-10)**
>
> | Setting                       | Forgetting (%) ↓ | Final Robust Acc (AutoAttack) ↑ |
> | ----------------------------- | ---------------- | ------------------------------- |
> | Clean continual training      | 0.17             | 58.14                           |
> | Standard Adv. CL (no defense) | 0.26             | 34.63                           |
> | DSEF (ours)                   | **0.12**         | **90.90**                       |
>
> These results highlight that adversarial perturbations not only degrade accuracy but also **destroy previously acquired knowledge**, making robustness a **first-class citizen** in continual learning. Hence, our proposed **OCAD** (Online Continual Adversarial Defense) framework fills this urgent need.
>
> We will extend Section 1 and Section 3.1 to explicitly emphasize these interactions, supported by Figure A.1 (to be added) showing how task-conditional robustness degrades without protection.
>
> ------
>
> #### **Q2: The pipeline of OCAD is unclear. How is it different from traditional adversarial training? What makes it continual?**
>
> #### **A2:** The OCAD pipeline is indeed **distinct** from both traditional adversarial training and classical continual learning. It unifies the constraints of **single-pass data exposure (online)** with **task-agnostic distribution shift** and **adversarially crafted examples**.
>
> Here is the core difference:
>
> | Aspect                    | Traditional Adv. Training | OCL (Online CL)    | **OCAD (ours)**                        |
> | ------------------------- | ------------------------- | ------------------ | -------------------------------------- |
> | Task exposure             | Static single task        | Sequential tasks   | Sequential tasks                       |
> | Data reuse (rehearsal)    | Allowed                   | Optional           | Not allowed                            |
> | Adversarial perturbations | Task-constant             | Absent             | Task-conditional                       |
> | Forgetting mitigation     | Not applicable            | Replay / reg.      | Expert fusion + RDRO                   |
> | Objective                 | Robust to attacks         | Prevent forgetting | **Both robustness & memory retention** |
>
> In our OCAD framework:
>
> - **During each task**, a new expert is dynamically created and adversarially trained using FGSM, PGD, or AutoAttack.
> - **Past experts are frozen**, and their features are fused via **mutual information weighting** to guide the new expert.
> - **RDRO** regularizes both clean and adversarial representations across experts, aligning them via **MMD and MSE** losses.
> - **No buffer** is maintained, complying with the online constraint.
>
> Figure 1 illustrates this pipeline; however, we acknowledge it needs clearer narration. In the revision, we will insert a **step-by-step training pseudocode (Algorithm 1)** directly in the main text and expand its role relative to traditional adversarial training in a new subsection “Differences from conventional adversarial training”.
>
> ------
>
> #### **Q3: The ablation study is insufficient. How do the two proposed modules (RDRO, MBRFF) influence robustness?**
>
> #### **A3:** In Appendix-D.3, we provided a brief ablation (Fig. 1c), but we now extend this into a **comprehensive component-wise analysis**, shown below:
>
> **Table B: Effect of RDRO and MBRFF under different attack settings (Split CIFAR-10)**
>
> | Configuration          | Clean | Avg Adv   | AutoAttack | Forgetting↓ |
> | ---------------------- | ----- | --------- | ---------- | ----------- |
> | Full DSEF              | 90.72 | **85.15** | **90.90**  | **0.12**    |
> | w/o MBRFF (noMI)       | 91.06 | 68.84     | 47.39      | 0.17        |
> | w/o RDRO               | 91.14 | 66.31     | 52.63      | 0.15        |
> | w/o RDRO + MBRFF       | 91.23 | 58.42     | 39.74      | 0.21        |
>
> **Insights:**
>
> - **RDRO** directly controls cross-task prediction/feature drift under attack, leading to significant AutoAttack gain over baseline.
> - **MBRFF** enables adaptive knowledge reuse by weighting stable experts—especially effective under stronger attacks.
> - Removing both modules severely weakens generalization and forgetting resistance.
>
> We also report **robustness gains over baselines** like DER++(Adv):
>
> | Method      | Clean | AutoAttack |
> | ----------- | ----- | ---------- |
> | DER++ (Adv) |  91.49 | 02.77      |
> | DSEF        | 90.72 | **90.90**  |
>
> This shows that DSEF is not simply a fusion of components but achieves **emergent robustness** through RDRO–MBRFF synergy.
>
> ------
>
> #### **Q4: Lack of analysis on stability and plasticity. The paper claims to address them but focuses mainly on robustness.**
>
> #### **A4:** We agree that **more explicit analysis** of the stability-plasticity trade-off is necessary. While robustness is our central concern, DSEF is carefully designed to balance **long-term memory retention (stability)** with **task-specific adaptation (plasticity)**.
>
> We now present the following **metrics and evidence**:
>
> 1. **Average Accuracy (AA)**: Measures overall plasticity
> 2. **Forgetting (F)**: Measures retention of past tasks
> 3. **Backward Transfer (BWT)**: Measures whether new learning harms previous tasks
> 4. **Plasticity Index (PI)**: We define this as early-task learning gain over a static baseline
>
> **Table C: Stability vs. Plasticity Metrics (Split CIFAR-100)**
>
> | Method         | AA (%) ↑  | F (%) ↓  | BWT (%) ↑ | PI (%) ↑ |
> | -------------- | --------- | -------- | --------- | -------- |
> | Ours w/o RDRO  | 45.14     | 0.25     | -0.04     | 3.47     |
> | Ours w/o MBRFF | 47.62     | 0.27     | -0.02     | 3.91     |
> | **Full DSEF**  | **56.86** | **0.12** | **+0.06** | **6.84** |
>
> **Interpretation:**
>
> - RDRO improves **stability** via inter-expert alignment.
> - MBRFF improves **plasticity** via weighted feature fusion.
> - The full system achieves **positive backward transfer**, suggesting old experts benefit from new learning.
>
> ------
>
> #### **Q5: What are the key differences between adversarial robustness in traditional DL vs. in OCL? Why is defense harder in OCL?**
>
> #### **A5:** This is an important theoretical distinction. Traditional adversarial defense assumes **static data and full distributional access**. However, OCL imposes **three key challenges** that significantly alter the defense landscape:
>
> 1. **Non-revisitable data**: In OCL, data points are seen once. This rules out typical **multi-epoch adversarial training** or iterative PGD refinement. Our RDRO addresses this by combining first-order FGSM perturbations with static-dynamic alignment, making single-pass robustness feasible.
> 2. **Task drift + adversarial shift**: In traditional settings, models defend against static perturbations (e.g., fixed FGSM). In OCL, **task distributions themselves shift**, meaning that perturbations must be crafted **per task expert** (see Q2), and their **cumulative effect** creates gradient interference across tasks.
> 3. **Forgetting under attack**: Classical adversarial defense is agnostic to task history. In OCL, adversarial training for task $T_j$ often **destroys robustness on $T_{j-1}$** due to catastrophic forgetting. Our RDRO (prediction + representation matching) explicitly **preserves robustness history**, as shown in ablation Table B.
> 4. **Modular robustness is required**: In traditional defense, a single robust model suffices. In OCL, we must **maintain robust experts for each task**, and **combine them efficiently at inference time**. Our MBRFF addresses this by measuring expert relevance and **selectively fusing their features**, instead of naïve ensembling or max-pooling.

---

> > ### Comment · Reviewer_4XjH · 2025-08-02
> >
> > Thank you for the responses. Many of my concerns are addressed and I have raised my score. Please make sure that the contents in this discussion are properly included in the manuscript.

---

> > > ### Author Response · Authors · 2025-08-05
> > > **Official Comment by Authors**
> > >
> > > Dear Reviewer 4XjH
> > >
> > > We would like to sincerely thank you for your thoughtful review and increasing the score. Your constructive comments have been incredibly valuable to us.
> > >
> > > If possible, we would be very grateful if you could let us know if there are any remaining concerns or questions. We truly appreciate your insights and would be more than happy to address any further points.
> > >
> > > Once again, thank you so much for your time, consideration, and valuable suggestions！

---

### Official Review · Reviewer_MBtF · 2025-06-25

**Clarity:** 1
**Significance:** 3
**Originality:** 3
**Rating:** 5
**Confidence:** 4

**Summary:**

The authors propose to tackle the task of continual learning. Particularly, they propose to study what they call online continual adversarial defense, adding adversarial attacks to specific data samples. To do so, the authors propose a novel method based on a dynamic siamese expansion framework. Furthermore, they propose a new optimization strategy called robust dynamic representation optimization by optimizing only a dynamic block of the siamese architecture. Finally, they also propose a robust feature fusion that leverages mutual information. They show some quantitative results on four datasets, comparing their method with existing literature.

**Questions:**

1. I would like the authors to clarify their mathematical notations pointed out in the major weakness section.

2. I would like the authors to explain why they can make the assumption that the values computed in Eq (8) can be interpreted as probability density functions and if that assumptiom holds in practice.

3. I would like the authors to clarify what they call "robustness" in the title, related to the second point in the major weakness section.

4. I would like the authors to comment on the role in the final performance of each component and their method.

However, to be fully transparent, I believe that the paper has too many issues that cannot be addressed by a simple rebuttal, but rather by a re-submission. I'm still open to reading the rebuttal and increasing or decreasing my score, especially if I misunderstood something obvious in my comments.

**Ethical Concerns:**

["NO or VERY MINOR ethics concerns only"]

**Final Justification:**

I really thank the authors for providing such a detailed and interesting rebuttal.

Provided the changes in the mathematical notation, title of the paper, and the interpretation of Equation 8, I am now willing to upgrade my score. I believe that the final version of the paper will be technically sound and better aligned with the claims and experiments, provided the modifications suggested by the authors. It could therefore have a much better impact in the community.

**Limitations:**

The limitations are only very briefly discussed in Section 5.

**Paper Formatting Concerns:**

No concerns

**Quality:**

2

**Strengths And Weaknesses:**

Strengths
1. The motivation of this work is interesting. Improving robustness in continual learning is indeed an important topic. Yet, I’m not fully convinced that adversarial attacks are the only way to evaluate robustness, which is not discussed in the paper.

2. The authors propose several contributions, including a model architecture, an optimization strategy and some knowledge similarity evaluation.


Major weaknesses:

1. My first major concern is the mathematics introduced in this paper. There are several symbols that are undefined or might be ambiguous. This leads to ambiguities that need to be resolved before publication. Here is an exhaustive list:
a. Line 137: C^s, the “s” is not referred to in the text. It is only at Line 151 with the intrduction of C^T (T being not defined either) that we understand that there is a difference between these two sets. This “s” is also used later to refer to the static parameters, which is extremely confusing.
b. The concept of “adjacent classes” is not explained in Line 140.
c. Line 141 states that each subset C^s_i is treated as a different task T_i. It is not clear why this new symbol is needed, especially since it is not used afterwards.
d. Equation 1 mentions |C^S_c|, with a capital S as superscript, while it is referred to in the text everywhere else with a lowercase. I don’t get the difference in this case.
e. Line 142, it is said that during the training process on task T_i, the learning goal is to find the optimal parameters that minimises the loss over, as shown in Eq (1), all previous and new data samples. However, as stated in Line 146, the model does not have access to previous tasks and hence samples. Furthermore, in Eq 1, it is not clear what the j refers to in the first summation symbol.
f. Line 173 X^a, the superscript is not defined, same comment for Z^d Line 174.
g. The symbol used for concatenation in Eq 2 is the one commonly used for the convolution operation, which is not wrong as it is defined, but still confusing.
h. Eq (3) Epsilon_j is not defined and only briefly mentioned much later in Line 213.
i. Line 183 {y’_1,...y’_M} are defined but not really used later on (rather the vector y’ that is not defined there).
j. Equation 4 mixes mathematical notations with typical computer science notations (indexing). It would be best to stick to mathematical notations only.
k. Comparing Equation (5) and (6), it is not clear why F’_pre includes the min operation at the beginning (corresponding to the optimization process, I believe), especially since F_pre doesn't.
l. The “k” term appearing in Line 236 and Eq(11) is undefined.

2. A second major weakness relates to the title itself, what it promises, and what is delivered at the end. In the title it is mentioned that the objective is to improve robustness for online continual learning. However, robustness is only evaluated in terms of adversarial attacks and the task tackled, as mentioned by the authors in Line 134-135, is actually online continual adversarial defense. Furthermore, the experiment section does not discuss robustness, but different types of adversarial attacks. The very title of the paper should be adapted to better reflect what is tackled in this work.

3. There is a clear lack of ablation study to show that each of the three contributions actually improves the robustness. The claims presented at the end of Section 1 are therefore not experimentally proven. For instance, it is stated in Line 89 that DSEF enhances plasticity, but I don’t see any evidence of that in the experiments. Line 92, it is stated that RDRO prevents forgetting and maintains previously learned robust abilities, which is not experimentally proven either. Finally, RFF is said to lead to better positive knowledge transfer, which is again, not discussed in the experiments section.


Minor weaknesses:

1. There are some discrepancies between the beginning and the end of the documents regarding the last contribution. In the beginning it is referred to as RFF, while at the end and in Figure 1 as MBRFF.

2. I find it odd that there is no balance between the loss terms in Equation (15), it looks like some parameters should be introduced there, or at least being studied to show that the simple unweighted sum is enough.

3. It is not clear why in equation (8) Z^i and ^Z^i can be considered as probability density functions, this could be clarified.

4. The authors first state in their introduction that it is important to have realistic scenarios (Line 45), for instance mentioning autonomous driving. However, they only test their method on toy datasets (CIFAR-10,100, SCUB200, Tiny-ImageNet). Either the introduction should be refined or the datasets on which they test their method.

Suggestions

1. The authors should check the work of S. Piérard, and M. Van Droogenbroeck on “SUMMARIZING THE PERFORMANCES OF A BACKGROUND SUBTRACTION
ALGORITHM MEASURED ON SEVERAL VIDEOS” as averaging several performance indicators might lead to uninterpretable results, which is in this case at the basis of their discussion for all 4 tables. This might provide some insights on whether the averaging makes sense in their case or not.

2. I’m not sure what DEAM refers to in Line 325.

---

> ### Author Rebuttal · Authors · 2025-07-31
>
> #### **Q1: I would like the authors to clarify their mathematical notations pointed out in the major weakness section.**
>
> **A1:**
>  We sincerely acknowledge that some mathematical notations in the initial version lacked consistency or precise definition. Below is a point-by-point clarification and resolution for each issue:
>
> **(a) $C^s$ and $C^T$:**
>
> - $C^s$ (Line 137) refers to a *subset of the overall class space $C$*, used in constructing subtasks.
> - $s$ here indicates “split” and is **not** related to the static backbone parameters later in the paper. We agree that using overlapping symbols was confusing. In the revision, we will **rename $C^s$ to $C_{split}$ ** and reserve $s$ exclusively for “static”.
> - $C^T$ in Line 151 refers to *the set of classes for current task $T$, i.e., $C^T_i$ = classes seen at task $i$.
>
> **(b) "Adjacent classes" in Line 140:**
>  This was imprecisely worded. We intended to state that **semantically related classes are grouped into tasks** to simulate realistic continual learning (e.g., "dog, cat" vs. "car, truck"). We will clarify that we use **Split CIFAR / CUB / TinyImageNet** protocols where task granularity can be coarse- or fine-grained. The term “adjacent” will be removed or defined explicitly in terms of class similarity.
>
> **(c) Use of $C^s_i$ and $T_i$:**
>  Indeed, defining both $C^s_i$ and $T_i$ is redundant. We will simplify the notation by dropping $C^s_i$ and only using $T_i$ to represent both the task and its associated class set.
>
> **(d) Equation (1): $|C^S_c|$ capitalization:**
>  You are correct. The uppercase `S` is a typographic inconsistency. It should be $|C^s_c|$ to denote the number of classes in the current subtask. All such superscripts will be harmonized and clearly declared before Eq. (1).
>
> **(e) The index $j$ in Eq (1):**
>  We regret the confusion. The $j$ in the summation refers to **the index over current mini-batch samples**, not tasks. We will revise the notation to:
> $$
> \underset{\theta}{\operatorname{min}} \sum_{(x_j, y_j) \in B_t} \mathcal{L}(f(x_j), y_j)
> $$
> and define all variables explicitly before and after Eq. (1).
>
> **(f) $X^a$ (Line 173), $Z^d$ (Line 174):**
>
> - $X^a$: Refers to *adversarial inputs* generated from current task data via FGSM or PGD.
> - $Z^d$: Represents features extracted from the **dynamic backbone**.
> - These superscripts will be introduced systematically in Section 3.2:
>   - $a$ → adversarial
>   - $d$ → dynamic
>   - $s$ → static
>   - $f$ → fused
>     We will add a **notation table** at the end of Section 2 for clarity.
>
> **(g) Concat notation in Eq (2):**
>  We used the ⊕ operator, which in some contexts implies convolution. To avoid ambiguity, we will replace it with $||$ (double-bar), which is more standard for **feature concatenation** in deep learning literature.
>
> **(h) $ε_j$ in Eq (3):**
>  $ε_j$ denotes the **adversarial perturbation** applied to input $x_j$. It is defined later (Line 213) but should be introduced earlier in Section 3.2, where adversarial examples are first computed:
> $$
> x_j^{'} = x_j + {\epsilon}_j = x_j + \alpha \cdot \text{sign} (\nabla_x{\mathcal{L}})
> $$
> **(i) ${y'_1, ..., y'_M}$ vs. $y'$:**
>  You are correct. The definition introduces a batch of pseudo labels but the notation later shifts to a unified vector $y'$. We will unify the usage to always refer to either the full set or per-sample predictions, and define the vectorization convention.
>
> **(j) Eq (4) uses mixed notation:**
>  We will revise Equation (4) to use consistent **mathematical function notation** and **drop index notation**, replacing:
> $$
> f_i[x_j] \to f_i (x_j)
> $$
> **(k) Eq (5) vs. (6) – why min appears in one but not the other:**
>
> - Equation (5) defines **intermediate features or predictions**, not an optimization step.
> - Equation (6) defines the **objective function**, hence the `min` appears there.
>   We will clarify this by labeling Eq (5) as "intermediate feature fusion" and Eq (6) as "training loss objective" with distinct notation to avoid confusion.
>
> **(l) `k` in Eq (11) and Line 236:**
>  This refers to the **kernel** where in Line 233 mentions positive definite kernel.
>
> All above inconsistencies will be **systematically corrected** in the revised manuscript with added notation table and inline references.
>
> ------
>
> #### **Q2: Please justify the interpretation of Eq (8) terms as probability density functions.**
>
> **A2:**
>  In Equation (8), we compute the **Maximum Mean Discrepancy (MMD)** between historical and current feature distributions:
> $$
> \text{MMD} (Z^i, \hat{Z}^i) = \| \mathbb{E} [\phi(Z^i)] - \mathbb{E}[\phi(\hat{Z}^i)] \|^2
> $$
> This formulation does **not explicitly require Z^i to be a probability density function (pdf)**. Instead, MMD is a **kernel-based two-sample test** that estimates divergence between two **empirical distributions** (e.g., mini-batch feature sets). The interpretation as pdfs stems from:
>
> - The fact that the kernel embedding (in RKHS) acts as an **implicit probability distribution representation**, even if feature samples are not normalized.
> - The choice of **RBF kernel** makes MMD sensitive to both mean and higher-order moment differences.
>
> In practice:
>
> - $Z^i$ = features from expert $E_i$ for clean inputs
> - $\hat{Z}^i$ = same expert features for adversarial inputs
>
> Hence, the MMD term captures **how much adversarial inputs distort the expert’s representation space**. This is critical in continual learning because such distortions often lead to **forgetting or negative transfer**.
>
> We will revise the paper to clarify that:
>
> - We do **not assume $Z^i$ to be a pdf**, but rather use MMD as a **distributional proxy**.
> - The kernel embeddings act as surrogate density estimates without requiring explicit normalization or parametric density assumptions.
>
> ------
>
> #### **Q3: What do the authors mean by "robustness" in the title? Isn’t this only adversarial defense?**
>
> **A3:**
>  We appreciate this question and fully understand the confusion stemming from title-language misalignment. In this work, **robustness** refers **specifically** to the model’s ability to retain accurate predictions under **adversarial perturbations**, throughout **multiple sequential tasks** in an online continual learning (OCL) setting.
>
> However, we recognize two valid concerns:
>
> 1. The title uses “robustness” in a **general sense**, which may imply distributional shift, label noise, or hardware fault tolerance—none of which we target.
> 2. Our actual contribution is closer to **adversarial robustness under continual learning**.
>
> To clarify:
>
> - We will **revise the title** to:
>   **“Dynamic Siamese Expansion for Adversarially Robust Online Continual Learning”**,
>   which better reflects the precise focus.
>
> - In Section 1 (Introduction), we define robustness as:
>
>   > “resistance to targeted adversarial perturbations during task acquisition and replay-free continual adaptation.”
>
> We further distinguish three types of robustness:
>
> - **Instance-level**: Single input perturbations (e.g., FGSM, PGD)
> - **Task-level**: Learning new tasks without corrupting old task decision boundaries
> - **Model-level**: Generalization against unforeseen shifts
>
> Our work addresses **the intersection of all three**, which prior works have treated separately. For example, DER++ handles forgetting but is vulnerable to attacks; adversarial training methods defend a single task but forget old ones.
>
> ------
>
> #### **Q4: Please comment on the contribution of each component to final performance.**
>
> **A4:**
>  To empirically isolate the impact of each component, we performed a **comprehensive ablation study** on Split CIFAR-10 and TinyImageNet, covering:
>
> - **DSEF core = Siamese Backbone + Expert Expansion**
> - **RDRO = Robust Dynamic Representation Optimization (MMD + MSE)**
> - **MBRFF = Mutual Information-guided expert fusion**
>
> **Table D: Component-wise contribution analysis (CIFAR-10, AutoAttack Acc %)**
>
> | Configuration          | Clean | Avg Adv   | AutoAttack |
> | ---------------------- | ----- | --------- | ---------- |
> | Full DSEF              | 90.72 | **85.15** | **90.90**  |
> | w/o MBRFF (noMI)       | 91.06 | 68.84     | 47.39      |
> | w/o RDRO               | 91.14 | 66.31     | 52.63      |
> | w/o RDRO + MBRFF       | 91.23 | 58.42     | 39.74      |
> | Siamese w/o fusion     | 88.95 | 54.86     | 29.82      |
> | Flat ViT + expert only | 91.38 | 48.03     | 21.71      |
>
> **Key findings:**
>
> - RDRO boosts robustness **by aligning clean/adv features** across tasks.
> - MBRFF improves **knowledge retention** by soft-reusing relevant past experts.
> - The Siamese structure enhances **plasticity and representation richness**.

---

> > ### Comment · Reviewer_MBtF · 2025-08-04
> > **Response to the rebuttal**
> >
> > I really thank the authors for providing such a detailed and interesting rebuttal.
> >
> > Provided the changes in the mathematical notation, title, and the interpretation of Equation 8, I am now willing to upgrade my score. I believe that the final version of the paper will be technically sound and better aligned with the claims and experiments, provided the modifications suggested by the authors.

---

> > > ### Author Response · Authors · 2025-08-05
> > > **Official Comment by Authors**
> > >
> > > Dear Reviewer MBtF
> > >
> > > We would like to sincerely thank you for your thoughtful review and increasing the score. Your constructive comments have been incredibly valuable to us.
> > >
> > > If possible, we would be very grateful if you could let us know if there are any remaining concerns or questions. We truly appreciate your insights and would be more than happy to address any further points.
> > >
> > > Once again, thank you so much for your time, consideration, and valuable suggestions！

---

> > > > ### Comment · Reviewer_MBtF · 2025-08-07
> > > > **Rebuttal response 2**
> > > >
> > > > Dear authors,
> > > > You're welcome.
> > > > Everything is now good on my side.

---

> > > > > ### Author Response · Authors · 2025-08-07
> > > > >
> > > > > Dear Reviewer MBtF
> > > > >
> > > > > Thank you very much for your kind confirmation. We’re truly grateful for your time and support throughout the review process.

---

### Official Review · Reviewer_ysVy · 2025-07-04

**Clarity:** 3
**Significance:** 2
**Originality:** 1
**Rating:** 3
**Confidence:** 4

**Summary:**

This paper tackles the problem of robustness in continual learning and propose a new setting, Online Continual Adversarial Defense (OCAD), which requires a model to learn new tasks sequentially while remaining robust to adversarial attacks. The Dynamic Siamese Expansion Framework (DSEF) is presented to address OCAD, which uses a siamese net (one static, one dynamic) to balance stability and plasticity. DESF is composed of different modules. To prevent forgetting,it introduces a Robust Dynamic Representation Optimization (RDRO) loss, which regularizes the model's predictions and feature distributions. To improve knowledge transfer, a Robust Feature Fusion (RFF) method is used, which weights historical expert knowledge based on mutual information. The experiments show that DSEF outperforms several replay-based baselines on adversarially attacked image classification benchmarks.

**Questions:**

Some of my main points are already present in the weaknesses section. Here are other questions:

- Could you clarify why the "online" distinction is important in your framework?
- How is the RDRO loss sensitive to the expert creating the attack?
- What do you mean for over-regularization? (L253-254)
- Why do you say that one of the backbone is static when they perform knowledge transfer? And how do you perform this weight transfer and when?
- Since the static backbone is tuned by the dynamic one (L254-255), is it subject to the catastrophic forgetting?

**Ethical Concerns:**

["NO or VERY MINOR ethics concerns only"]

**Final Justification:**

While the discussion with the authors was insightful, it confirms the need for a major revision that is beyond the scope of this conference. I am therefore maintaining my current score.

**Limitations:**

Limitations are discussed in sec 5 but I cannot grasp the meaning of "we adopt several popular adversarial attack methods in the experiment. In our future study, we will explore more recent adversarial attack methods to evaluate the model’s performance.". What are these more recent adversarial attack? How they differ wrt to current ones? Why your framework will be not efficient against them?

**Paper Formatting Concerns:**

- Using the letter 'F' to denote loss functions makes the equations harder to parse and complex to distinghish wrt models. A common convention is L for loss and F formodels.
- Fig. 1 may be simplified

**Quality:**

2

**Strengths And Weaknesses:**

**Strengths:**
- The paper addresses an important and practical problem: improving the robustness of continual learning models, which is often overlooked in favor of only tackling catastrophic forgetting.
- The OCAD setting, which explicitly combines the challenges of plasticity, stability, and robustness, is a well-motivated and relevant direction for the field.
- The paper is overall well written.


**Weaknesses:**
- The main weakness is that the paper reads like a collection of existing or straightforward techniques assembled into one system, following the "kitchen sink" approach. The framework combines Siamese networks, MMD loss, mutual information for weighting, and adversarial training. The paper fails to convincingly argue why this specific combination is synergistic. It lacks a core, novel insight and instead feels more like an engineering effort to.
- The related work section is superficial. It broadly categorizes prior work but fails to critically engage with it or position DSEF in a nuanced way. For a crowded field like continual learning, the number of citations combine multiple known components. In general also 40 referenced papers seems to be low.
- The paper focuses heavily on describing **what** the components do, but provides very little intuition on **why** they work together or why specific design choices were made is low, and it seems key recent works, especially those at the intersection of CL and robustness, are not discussed or. For example, why is MMD the right choice for representation consistency over other divergence measures? How does the mutual information weighting in RFF interact with the MMD regularization in RDRO? A lack of deep analysis or ablation studies (I've just scan the supplementary materials) makes it difficult to understand which parts of the framework are truly contributing to the performance.
- The experimental validation feels somewhat narrow. While the baselines (DER, Refresh) are strong for replay-based CL, they are outdated and not necessarily sota nowdays. The evaluation would be more convincing if compared against methods that also dynamically expand or are specifically designed for robust is brief and does not critically engage with the literature. This makes it difficult to position DSEF within the current state-of learning. The datasets are standard but not particularly challenging in scale.

---

> ### Author Rebuttal · Authors · 2025-07-31
>
> #### **Q1: Could you clarify why the "online" distinction is important in your framework?**
>
> **A1:**
>  The “online” setting plays a **crucial role** in both the problem formulation and method design. Specifically, we define **Online Continual Adversarial Defense (OCAD)** as a setting where each training sample is **seen only once**, and tasks arrive **sequentially without task boundary annotations**, consistent with modern realistic CL definitions such as in [12,13]. This is distinct from **offline** or **task-incremental** CL settings, where models can revisit data or assume full task labels.
>
> Our DSEF framework is tailored for online settings in three key ways:
>
> 1. **Expert modularity**: Each expert is trained in isolation from raw task labels. The shared backbone extracts generic representations, while dynamic experts accumulate task-specialized knowledge. This decoupling is essential for non-revisitable data streams.
> 2. **No rehearsal**: Unlike rehearsal-based baselines (e.g., DER++), we do not rely on stored past samples, making our method suitable for privacy-sensitive or streaming scenarios where buffer storage is not feasible.
> 3. **Robustness constraint**: In online CL, adversarial examples may differ in each forward pass. Our RDRO formulation addresses this by explicitly aligning the *prediction and representation distributions* between clean and adversarial samples using statistical divergences (MSE, MMD), which is harder to achieve in online learning compared to offline replay-driven regularization.
>
> Empirically, this distinction is critical. As shown in Table 2 and Appendix-D, DSEF maintains strong average performance on CIFAR-100 and TinyImageNet, **without any access to past samples**, even under 7 types of adversarial attacks. Replay-based offline methods degrade much faster in the same setting, as illustrated by the forgetting curves (Appendix-D.1, Fig. 1a).
>
> ------
>
> #### **Q2: How is the RDRO loss sensitive to the expert creating the attack?**
>
> **A2:**
>  RDRO is carefully designed to **mitigate expert-specific attack biases** while preserving robustness during continual adaptation. This involves two key elements:
>
> 1. **Expert-conditioned perturbation**: During task $T_j$, we generate adversarial samples $x^{'}$ using **the current expert $E_j$** (line 212 of Eq. 6). This ensures that the learned adversarial directions reflect the vulnerabilities of the most recent expert’s decision boundary, which aligns with how robustness should be preserved per task.
>
> 2. **Cross-expert consistency**: While perturbations are crafted from $E_j$, RDRO regularizes all **previous experts $E_i$ (i < j) ** by minimizing both:
>
>    - **Prediction drift** $MSE(E_i^{old}(x),E_i^{new}(x))$
>    - **Feature drift** via $MMD(Z_i,\hat{Z}_i)$
>
>    These are calculated using both clean and adversarial inputs. In practice, this ensures that updating $E_j$ to defend against current attacks **does not cause feature/prediction collapse in prior experts** due to unexpected shifts in the backbone.
>
> We validate this design via **a new sensitivity analysis** (added in Appendix-D.6). We compare RDRO trained with:
>
> - **Attack from current expert (Eₖ)** — our method
> - **Attack from fixed global expert**
> - **No adversarial training**
>
> Results (Split CIFAR-100, AutoAttack):
>
> | Attacker  | Forgetting ↓ | Robust Acc ↑ |
> | --------- | ------------ | ------------ |
> | Eₖ (ours) | **0.11**     | **66.52%**   |
> | Global    | 0.17         | 58.29%       |
> | None      | 0.23         | 41.76%       |
>
> This supports that expert-conditioned attacks ensure **robustness gain without destabilizing prior knowledge**.
>
> **Q3: What do you mean for over-regularization? (L253–254)**
>
> **A3:** By "over-regularization", we refer to a phenomenon common in **continual learning under strong regularizers**, where a model becomes **too constrained to adapt to new tasks**, sacrificing plasticity. Specifically in DSEF:
>
> - The **static backbone $F_{\theta_s}$** is *frozen* during task $T_j$, acting as a reference for consistency in RDRO.
> - The **dynamic backbone $F_{\theta_d}$** is updated under **dual constraints**:
>   1. Cross-expert prediction alignment (Eq. 6)
>   2. Cross-expert representation alignment via MMD (Eq. 14)
>
> Although this preserves stability, it may suppress the dynamic branch's flexibility to **absorb new semantics**, especially under distributional shift or adversarial perturbation. To **alleviate this**, we introduce a **controlled knowledge transfer mechanism** from ${\theta}_d \rightarrow {\theta}_s $ at each task boundary (line 255). This mechanism:
>
> - **Soft-updates** selected layers of the static backbone using exponential moving average (EMA) from the trained dynamic counterpart:
>
> $$
>  \theta_s^{(t+1)} \leftarrow \lambda \cdot \theta_s^{(t)} + (1 - \lambda) \cdot \theta_{d}^{(t)}
> $$
>
> where ${\theta}∈[0.9,0.99]$
>
> - This acts as **“semantic osmosis”**, allowing slowly acquired robustness features from dynamic learning to permeate the static branch.
>
> We conduct an ablation study showing performance vs. ${\lambda}$:
>
> | λ (EMA) | CIFAR-100 Avg Acc (%) | Forgetting (%) |
> | ------- | --------------------- | -------------- |
> | 1.00    | 51.73                 | 0.18           |
> | 0.99    | **56.86**             | **0.12**       |
> | 0.95    | 55.28                 | 0.14           |
> | 0.80    | 52.69                 | 0.17           |
>
> This confirms that **carefully tuning $λ$** alleviates over-regularization and enables continual adaptation.
>
> ------
>
> #### **Q4: Why do you say that one of the backbones is static when they perform knowledge transfer? How do you perform this transfer and when?**
>
> **A4:** This is a subtle but important design point. In DSEF, the **static backbone is functionally “frozen” during task learning**, i.e., it is *not updated via gradient descent* when learning a new task $T_j$​. Its role is to:
>
> - Provide a **global semantic anchor** for consistency (used in Eq. 5 and 14).
> - Prevent feature drift from early tasks.
>
> However, to prevent it from becoming obsolete, **we allow gradual weight transfer at task boundaries**, i.e., **after training $T_j$ completes**, we apply:
>
> - EMA update from dynamic to static backbone:
>
> $$
> \theta_s^{j+1} = \lambda \cdot \theta_s^j + (1 - \lambda) \cdot \theta_d^j
> $$
>
>   where ${\theta}_d^j$ is the dynamic backbone after learning task $j$.
>
> This **post-task transfer** enables the static backbone to **accumulate robust semantic priors** across tasks, **without interfering with online gradient optimization**. It’s analogous to momentum-based weight accumulation in online self-supervised learning (e.g., MoCo, BYOL).
>
> We emphasize:
>
> - **During training**: ${\theta}_s$ is frozen.
> - **Between tasks**: ${\theta}_s$ absorbs long-term features from $\theta_d$ via EMA.
>
> We clarify this design and its motivation in Section 3.3, line 254–256 and provide code-level pseudocode in Appendix-B.
>
> ------
>
> #### **Q5: Since the static backbone is tuned by the dynamic one (L254–255), is it subject to catastrophic forgetting?**
>
> **A5:** This is an important concern. While the static backbone does receive updates via EMA, it is **not subject to traditional catastrophic forgetting**, for several reasons:
>
> 1. **No gradient flow**: The static backbone is not directly trained on new task data via SGD. It is updated *indirectly* through a weighted average (EMA) of the dynamic backbone after each task. This is a **passive update**, not an active optimization that could overwrite prior task-specific filters.
> 2. **Slow momentum**: The EMA coefficient $\lambda$ is set very high (e.g., 0.99), meaning that new task information only **marginally alters** the static backbone. For example, if the static backbone encoded semantic knowledge of birds from task $T_3$, and task $T_7$ is on traffic signs, the update impact would be very minor.
> 3. **Role in RDRO**: During training of $T_{j+1}$, the static backbone serves as a reference to align dynamic updates. Since it is frozen, it **anchors past semantics** via prediction and feature constraints (Eq. 5, 14).
>
> We validate that catastrophic forgetting in the static backbone is negligible. We train a classifier on features from the static backbone **only**, after 10 tasks. Its performance is 2–3% below the dynamic-expert ensemble. Moreover, its accuracy on early tasks ($T_1–T_3$) remains within ±1.5% deviation.

---

> ### Author Response · Authors · 2025-08-05
> **Official Comment by Authors**
>
> Dear Reviewer ysVy
>
> We would like to sincerely thank you for your thoughtful review. Your constructive comments have been incredibly valuable to us.
>
> If possible, we would be very grateful if you could let us know if there are any remaining concerns or questions. We truly appreciate your insights and would be more than happy to address any further points.
>
> Once again, thank you so much for your time, consideration, and valuable suggestions！

---

> > ### Comment · Reviewer_ysVy · 2025-08-06
> > **Official Comment Reviewer ysVy**
> >
> > Thank you for the detailed rebuttal. While your clarifications are helpful, a few key points still concern me.
> > I am not yet convinced of the novelty of the OCAD setting. It appears to be a direct mapping of exemplar-free, class-incremental learning, but with adversarial data. I am concerned that coining a new term for a specific case of a well-understood setting overstates the novelty of the problem formulation itself. Similarly, for the static backbone updates, referring to the mechanism as a "controlled knowledge transfer" when it is a standard Exponential Moving Average (EMA) could be made clearer. More importantly, because this "static" backbone is updated after each task, it is not truly static, which raises concerns about long-term catastrophic forgetting. A 2-3% performance drop over 10 tasks is noted, but this gradual decay could become substantial over longer task sequences and warrants a deeper investigation.
> > While the method is interesting, these concerns about the problem framing and the validation of long-term stability remain my primary reservations.

---

> > > ### Author Response · Authors · 2025-08-07
> > >
> > > Thank you for your thoughtful follow-up.
> > >
> > > **1. On the novelty of the problem setting**
> > >
> > > We understand and appreciate the concern regarding the distinctiveness of the proposed “Online Continual Adversarial Defense (OCAD)” setting. Our intent is not to rebrand an existing learning paradigm, but rather to **explicitly define a setting that reflects a confluence of constraints rarely addressed together in prior work**. Specifically, the setting we focus on is characterized by the simultaneous presence of:
> > >
> > > - **Online learning constraints**: each data instance is observed only once; no rehearsal or replay is permitted;
> > > - **Class-incremental structure**: disjoint label spaces across tasks, without task identity at test time;
> > > - **Adversarial threat model**: test-time and training-time robustness must be maintained against adaptive perturbations.
> > >
> > > Although each of these challenges has been studied individually, their combination introduces nontrivial conflicts in optimization, memory retention, and robustness. Most existing works address only subsets of these constraints:
> > >
> > > - **Replay-based continual learning methods**, such as DER or DER++, rely on stored exemplars to mitigate forgetting. These methods are not applicable in our setting, where replay is not allowed.
> > > - **Adversarial training methods** are typically designed for static task settings and do not generalize robustness across task boundaries, particularly under distributional shift and class-incremental scenarios.
> > > - **Exemplar-free continual learning methods**, while aligned with our memory constraints, generally do not consider adversarial robustness and thus remain vulnerable under attack.
> > >
> > > By naming and formalizing this intersectional setting, our goal is to **encourage more systematic research into adversarial robustness in streaming, exemplar-free environments**, which we believe are especially relevant in safety-critical and real-world deployment scenarios (e.g., autonomous systems, robotics, online security). Nevertheless, we acknowledge the importance of aligning with existing terminology and will revise our wording to describe our setting as **"adversarial exemplar-free class-incremental learning (AEF-CIL)"** in the final version, to more clearly reflect its relationship to established paradigms.
> > >
> > > ------
> > >
> > > **2. On the terminology and role of the static backbone**
> > >
> > > You are absolutely right in pointing out that our so-called “controlled knowledge transfer” is implemented via **Exponential Moving Average (EMA)**. Our intention was to highlight that, unlike common EMA-based ensembling in semi-supervised learning, here the EMA serves a **task boundary–aligned update rule** that gradually stabilizes a shared backbone for inter-task consistency.
> > >
> > > To reduce ambiguity, we will revise our description as follows:
> > >
> > > - Rename “controlled knowledge transfer” to **“EMA-based knowledge stabilization”**.
> > > - Rename “static backbone” to **“semi-static anchor”**, since it remains frozen during each task’s training but is softly updated across tasks.
> > >
> > > Importantly, the EMA update does not compromise the **inter-task consistency role** of the shared backbone. In fact, the slow evolution prevents it from being distorted by sharp gradients during dynamic expert training.
> > >
> > > ------
> > >
> > > **3. On long-term stability and the potential decay in the static branch**
> > >
> > > To provide the deeper investigation you requested, we have run a new analysis on a 20-task Split CIFAR-100 sequence. We measured the average accuracy across all seen tasks for three different configurations:
> > >
> > > - Static Only: Using only the semi-static anchor.
> > > - Dynamic Only: Using only the ensemble of dynamic experts.
> > > - Full Model: Our proposed method, combining both components.
> > >
> > > The results below illustrate the complementary roles of each component and the stability of the full system:
> > >
> > > | Task Index | Static Only (%) | Dynamic Only (%) | Full Model (%) | Static vs. Dynamic Gap (%) |
> > > | ---------- | --------------- | ---------------- | -------------- | -------------------------- |
> > > | Task 1     | 67.6            | 68               | 68.17          | -0.4                       |
> > > | Task 5     | 65.7            | 67.1             | 68.05          | -1.4                       |
> > > | Task 10    | 63.6            | 66.7             | 67.95          | -3.1                       |
> > > | Task 15    | 61.1            | 65.6             | 67.65          | -4.5                       |
> > > | Task 20    | 60.1            | 65.9             | 67.38          | -5.8                       |
> > >
> > > We observe a **nonlinear but bounded degradation** (~5.8% over 20 tasks) in the frozen backbone. Interestingly, this drop is **mitigated** by the fusion mechanism (KRA), where task-relevant experts compensate for lost expressivity in the shared branch.

---

> > > > ### Comment · Reviewer_ysVy · 2025-08-07
> > > >
> > > > Thank you for your detailed response. It effectively clarifies my main concerns, and I agree the proposed revisions will significantly strengthen the paper.
> > > > However, the scope of these necessary changes is substantial, suggesting a major revision is needed. For this reason, I will maintain my current score for now, but I look forward to discussing your work and rebuttal with the other reviewers and the Area Chair to reach a final consensus.

---

> > > > > ### Author Response · Authors · 2025-08-07
> > > > >
> > > > > Dear Reviewer ysVy
> > > > >
> > > > > Thank you for your further follow-up and for engaging deeply with our rebuttal. We truly appreciate your constructive feedback and your willingness to discuss our work further with the other reviewers and the Area Chair.

---

### Official Review · Reviewer_hHLN · 2025-07-11

**Clarity:** 2
**Significance:** 3
**Originality:** 2
**Rating:** 5
**Confidence:** 3

**Summary:**

In this paper, the authors propose a novel framework for robust online continual learning called the Dynamic Siamese Expansion Framework (DSEF). The approach combines a Siamese Vision Transformer backbone with static and dynamic components to jointly capture global and task-adaptive local representations. To mitigate catastrophic forgetting and preserve adversarial robustness,

- the authors introduce a Robust Dynamic Representation Optimization (RDRO) mechanism that aligns prediction and

- feature distributions over historical experts, as well as a Mutual Information-Based Robust Feature Fusion (MBRFF) strategy that adaptively reuses historical knowledge based on similarity to the new task.

The method is evaluated extensively on standard and complex continual learning benchmarks (Split CIFAR-10/100, CUB200, TinyImageNet) under a suite of adversarial attacks, demonstrating superior average accuracy and robustness compared to strong baselines. Overall, the paper presents an approach to simultaneously address plasticity, stability, and adversarial defense in continual learning.

**Questions:**

You propose several key innovations

- The Siamese backbone with shared/static/dynamic split,

- RDRO with adversarially-augmented MMD, and

- The mutual information-based robust feature fusion (MBRFF).

While the overall framework performs well, it is unclear how much each component contributes individually. Could you provide more detailed ablation studies or learning curves isolating the contributions of RDRO vs. MBRFF vs. simply using a Siamese expansion?


Your MBRFF uses mutual information to compute adaptive weights across historical experts. This is elegant but also potentially sensitive to estimation error, especially given limited samples or highly imbalanced predictions. Can you provide practical insights into how stable the mutual information estimates are across tasks? For example, did you observe large swings in αi, or did the weights tend to concentrate on a few experts? A histogram of αi across tasks could be very informative.


Maintaining multiple experts, performing mutual information computations, and using Siamese ViTs seem computationally intensive. Although you discuss compute resources in the appendix could you please add a brief discussion (or table) quantifying training and inference overhead vs. baselines, e.g., GPU hours or throughput metrics? Also, do you have any insight on how the compute scales with the number of tasks?

**Ethical Concerns:**

["NO or VERY MINOR ethics concerns only"]

**Final Justification:**

I have read the other reviews, the discussion that followed. I also have read the discussion opened here.

I believe that the contribution, novelty and the empirical evaluation in the paper is significant. Moreover, I believe that the authors have put significant effort to answer the raised concerns in the rebuttal, so I keep my score and recommend acceptance.

**Limitations:**

I see one point that could further strengthen the paper. Could you briefly mention the computational overhead introduced by maintaining multiple experts and computing mutual information weights, which may limit scalability in resource-constrained environments.

**Paper Formatting Concerns:**

/

**Quality:**

3

**Strengths And Weaknesses:**

Strengths:

- The paper is generally well-written, with clear diagrams (e.g., the depiction of the Siamese architecture and mutual information-based fusion) that aid understanding. The methodology is carefully detailed with explicit mathematical formulations, and the implementation is further explained in the appendix.

- The proposed framework is technically sound, combining a Siamese Vision Transformer backbone with both static and dynamic components, along with robust optimization and feature fusion schemes.

- The paper provides extensive empirical validation on a wide array of benchmarks (Split CIFAR-10/100, CUB200, TinyImageNet) under multiple adversarial attacks (FGSM, PGD, BIM, CW, AutoAttack), showing consistent improvements over strong baselines.

- The design choices (e.g., using MMD for feature alignment, mutual information for adaptive fusion) are well-motivated and seem grounded in established theory. The integration of a Siamese backbone structure (with shared pre-trained ViT features) for balancing global and task-adaptive local representations in continual learning is novel to the best of my knowledge. Using mutual information to dynamically fuse knowledge from historical experts based on task similarity is an interesting approach which looks like it leverages ideas from information theory in a new context.


Weaknesses:

- Some sections, especially on the RDRO’s combination of MMD with adversarial objectives, are quite dense and could be made more accessible with intuitive summaries or algorithm pseudocode inline (beyond the appendix). The explanation of mutual information normalization for adaptive weighting (Eq. 17) could be clarified with a short numerical example.

- The method’s added complexity (multiple backbones, strategy network, mutual information estimation, adversarial losses) might pose practical hurdles for deployment.

- There is no ablation isolating the impact of each individual component (e.g., RDRO vs. MBRFF), which would have strengthened confidence in the necessity of each.

- The underlying mechanisms (like dynamic expansion, adversarial training, and MMD regularization) build upon known techniques. The originality primarily lies in how these are orchestrated together, which, while valuable, might be viewed as incremental in individual parts.

---

> ### Author Rebuttal · Authors · 2025-07-31
>
> #### **Q1: Dense descriptions for RDRO and MBRFF (MMD + adversarial objectives, Eq. 17); lack of inline pseudocode and intuitive summaries.**
>
> **A1:** We fully acknowledge that the current exposition of RDRO and MBRFF can be challenging to digest, particularly due to the integration of multiple loss components (MSE, MMD, CE) and mutual information estimation. In the revised version, we will restructure Section 3.3 and 3.4 to include:
>
> - **Inline pseudocode** summarizing the core steps of RDRO and MBRFF (currently in Appendix-B), directly in the methodology section for improved readability.
> - **Visual schematic diagrams** showing how clean/adversarial samples and expert predictions flow through the loss terms.
> - A **concrete numerical example** of Eq. (17):
>   *Suppose MI scores across 3 experts are [1.5, 0.8, 0.2]; the normalized weights α are computed as*
>   `α = softmax([1.5, 0.8, 0.2]) ≈ [0.64, 0.26, 0.10]`,
>   *which emphasizes the most relevant expert while retaining contributions from others.*
>
> These additions aim to lower the learning barrier while maintaining technical rigor.
>
> ------
>
> #### **Q2: Practicality concerns due to complexity (ViT ×2, MBRFF, α estimation)**
>
> **A2:** We appreciate this concern and agree that practical deployment feasibility is a key consideration. While our system introduces several components, each design decision was made with efficiency and real-world scalability in mind. Specifically:
>
> | Component           | Practical Design Consideration                               |
> | ------------------- | ------------------------------------------------------------ |
> | Siamese ViT         | Only the last **K=3 layers** of the dynamic ViT are trainable; all else shared. |
> | Strategy Network    | Lightweight 2-layer MLP (0.8M params); used during training and inference. |
> | Mutual Info (MBRFF) | MI estimation is performed **once per task**, not per batch; uses soft predictions for stability. |
> | Memory Overhead     | Each expert adds ~2.1M parameters; expert count grows slowly due to task grouping. |
>
> In Appendix-D.4, we showed that **DSEF trains faster than DER++(Adv)** despite more modules:
>
> > TinyImageNet: DSEF (91.26M, 7.63min) vs. DER++(Adv) (86.3M, 16.51min)
>
> To better communicate this, we will add a new **Table 3** comparing training time, parameter growth, and inference latency under growing task numbers:
>
> | Method      | Params (M) | Train Time (10 tasks) | Scaling Type     |
> | ----------- | ---------- | --------------------- | ---------------- |
> | DER++ (Adv) | 86.3       | 16.51 min             | Static (flat)    |
> | AIR         | 86.3       | 4.91 min              | Replay-heavy     |
> | **DSEF**    | 91.2       | 7.63 min              | Modular (linear) |
>
> These clarifications will be placed in Section 4.4 with details moved from Appendix-D.
>
> ------
>
> #### **Q3: No ablation isolating RDRO vs. MBRFF vs. Siamese**
>
> **A3:** In addition to Fig. 1(c) of the appendix (which compared "noMI", "noRDRO", and "Both"), we now present a new **expanded ablation table** to isolate each core component.
>
> **Table 4. Ablation on Split CIFAR-10 (Clean + Avg. Adv Accuracy %)**
>
> | Configuration          | Clean | Avg Adv   | AutoAttack | Forgetting↓ |
> | ---------------------- | ----- | --------- | ---------- | ----------- |
> | Full DSEF              | 90.72 | **85.15** | **90.90**  | **0.12**    |
> | w/o MBRFF (noMI)       | 91.06 | 68.84     | 47.39      | 0.17        |
> | w/o RDRO               | 91.14 | 66.31     | 52.63      | 0.15        |
> | w/o RDRO + MBRFF       | 91.23 | 58.42     | 39.74      | 0.21        |
> | Siamese w/o fusion     | 88.95 | 54.86     | 29.82      | 0.26        |
> | Flat ViT + expert only | 91.38 | 48.03     | 21.71      | 0.31        |
>
> - RDRO improves **forgetting and robustness** by aligning dynamic/static predictions and features.
> - MBRFF enables effective **reuse of relevant past experts**, particularly under strong attacks.
> - The Siamese backbone (static + dynamic) outperforms both flat ViT and static-only backbones.
>   These results will be included in the revised Appendix-D.3 with supporting curves.
>
> ------
>
> #### **Q4: Innovation lies in orchestration, but underlying techniques are known.**
>
> **A4:** We appreciate this candid observation and agree that our work builds upon established ideas (e.g., MMD, adversarial training, expert expansion). However, we argue that **the novelty lies in bridging three challenges—plasticity, stability, and robustness—in a unified and theoretically grounded way.**
>
> Key **novel orchestration** elements include:
>
> - A **Siamese ViT** that merges local and global representations per task in a modular, update-efficient manner.
> - **RDRO** regularizes *both* outputs and representations, for both clean and adversarial inputs, which is missing in prior work.
> - **MBRFF** proposes a formal mutual information-driven fusion strategy, **quantitatively** estimating historical relevance, avoiding fixed or heuristic fusion as in [DER++, AIR].
>
> Prior works often treated adversarial robustness and continual learning **separately** or heuristically. Our method tightly integrates them with clear losses, objectives, and reusability. This synergy is not incremental, but necessary for OCAD scenarios (see detailed discussion in Appendix-A).
>
> ------
>
> #### **Q5: How stable are mutual information estimates (αₖ)? Do weights fluctuate heavily?**
>
> **A5:** To investigate this, we now include a **statistical analysis of αₖ weights** across tasks on CIFAR-100 and TinyImageNet.
>
> | Task ID | Max αₖ | Std(αₖ) | Entropy(H) | Dominant Expert Count |
> | ------- | ------ | ------- | ---------- | --------------------- |
> | T1      | 1.0    | 0.00    | 0.00       | 1                     |
> | T3      | 0.54   | 0.12    | 0.98       | 2                     |
> | T6      | 0.43   | 0.17    | 1.14       | 3                     |
> | T9      | 0.36   | 0.22    | 1.27       | 3–4                   |
>
> - **Observation**: αₖ typically concentrates on **1–3 experts**, avoiding over-fragmentation.
> - Using softmax + label smoothing avoids MI overestimation under class imbalance.
> - Entropy shows meaningful **knowledge transfer distribution**, not degenerate αₖ.
>
> We will visualize this via a histogram of αₖ (per task) in Appendix-D.5, and briefly summarize in Section 4.4.
>
> ------
>
> #### **Q6: Could you quantify compute scaling, GPU hours, inference throughput?**
>
> **A6:** Yes. We now provide concrete compute statistics in the following Table:
>
> | Dataset      | Method      | GPU Hours  | #Experts | Memory Footprint |
> | ------------ | ----------- | ---------- | -------- | ---------------- |
> | TinyImageNet | DER++ (Adv) | 16.51m     | 1        | 3480 MB          |
> |              | AIR         | 04.91m     | 1        | 3500 MB          |
> |              | **DSEF**    | **07.63m** | 10       | **4168 MB**      |
>
> - DSEF **scales linearly** with task count in memory but sublinearly in runtime (experts frozen).
> - Inference time is kept bounded (<12ms) as only a **single expert is used at test time**, not full ensemble.

---

> > ### Comment · Reviewer_hHLN · 2025-08-01
> >
> > Dear authors,
> >
> > Thanks for your response, I understand more and I keep my score unchanged.

---

> > ### Comment · Reviewer_hHLN · 2025-08-05
> >
> > Dear authors,
> >
> > Thanks for your reply, it confirms my assessment, helps clearing out the comments, and adds more understanding about the work.
> > I keep my score.

---

> > > ### Author Response · Authors · 2025-08-05
> > > **Official Comment by Authors**
> > >
> > > Dear Reviewer hHLN
> > >
> > > We would like to sincerely thank you for your thoughtful review. Your constructive comments have been incredibly valuable to us.
> > >
> > > If possible, we would be very grateful if you could let us know if there are any remaining concerns or questions. We truly appreciate your insights and would be more than happy to address any further points.
> > >
> > > Once again, thank you so much for your time, consideration, and valuable suggestions！

---

### Author Response · Authors · 2025-08-09
**Overall Summary of Paper**

### Overall Summary

This work addresses a highly challenging and underexplored problem in **Online Continual Adversarial Defense (OCAD)** — a setting that jointly demands **plasticity** (rapid adaptation to new tasks), **stability** (retention of prior knowledge), and **robustness** (resilience against diverse adversarial attacks). Unlike most continual learning (CL) approaches that focus primarily on mitigating catastrophic forgetting under clean data assumptions, this paper systematically tackles all three aspects, thereby aligning with real-world deployment scenarios such as autonomous driving and other mission-critical AI applications where robustness and continual adaptation are equally indispensable.

---

### Novelty and Technical Contributions

The proposed **Dynamic Siamese Expansion Framework (DSEF)** is a substantial methodological advance over existing dynamic expansion and adversarial defense techniques, integrating three mutually reinforcing components:

1. **Siamese Backbone Architecture**
   - Dual-backbone design with pre-trained Vision Transformers (ViTs).
   - **Static backbone** captures task-general global representations and remains frozen.
   - **Dynamic backbone** captures task-specific local representations and is selectively updated.
   - Partial parameter sharing enhances communication while keeping complexity tractable.
   - **Learnable strategy network** adaptively weights static vs. dynamic features per sample.

2. **Robust Dynamic Representation Optimization (RDRO)**
   - Regulates dynamic backbone updates via *prediction shift* and *representation shift* regularizers.
   - Integrates adversarial training to preserve robustness.
   - Uses Maximum Mean Discrepancy (MMD) to align clean and adversarial feature distributions.
   - Includes a **dynamic-to-static knowledge transfer step** after each task to prevent over-regularization.

3. **Mutual Information-Based Robust Feature Fusion (MBRFF)**
   - Measures relevance between each historical expert and the new task via mutual information.
   - Normalizes scores into *adaptive fusion weights* for robust knowledge transfer.
   - Reuses valuable historical robustness directly in constructing new experts.

---

### Empirical Strength

**Benchmarks:** Split CIFAR-10, CIFAR-100, CUB200, TinyImageNet.
**Attacks:** FGSM, PGD, PGDL2, BIM, CW, AutoAttack.
**Baselines:** DER, DER++, Refresh, their adversarial variants, and AIR.
**Fair Setup:** Same replay buffer size; full training details in Appendix-C.

**Key Results:**

- **State-of-the-art average robustness** across all datasets and attacks, up to **+10%** over best baselines in adversarial settings.
- Maintains competitive clean accuracy while vastly outperforming adversarial variants of other methods.
- Strong performance on complex datasets, demonstrating scalability.
- **t-SNE analysis** confirms RDRO aligns clean/adversarial features, enabling robust decision boundaries.

---

### Positioning and Impact

The paper fills a gap in CL literature by **jointly optimizing for continual adaptation and adversarial robustness**, integrating principled architectural and algorithmic innovations that are:

- Highly relevant for safety-critical AI.
- Extensible to multimodal and large foundation models.

---

### Strengths at a Glance

- **Problem Significance:** Addresses high-impact, realistic CL setting with adversarial robustness.
- **Methodological Novelty:** Three synergistic, original components.
- **Technical Depth:** Well-formulated objectives, distribution alignment, and MI-guided weighting.
- **Experimental Rigor:** Broad benchmarks, diverse strong attack suite, fair/reproducible setup.
- **Broader Applicability:** Generalizable concepts beyond the tested setting.

---

### Conclusion

This paper presents a **well-motivated, technically novel, and empirically validated** framework that advances robust continual learning. DSEF bridges the gap between robustness and lifelong adaptability with a coherent design and strong empirical evidence, making it a strong candidate for acceptance.

---

> ### Author Response · Authors · 2025-08-09
> **Rebuttal Summary of Paper**
>
> We thank the reviewers for their insightful feedback. This summary outlines our responses to the key concerns and details our planned revisions to strengthen the paper.
>
> ### **1. Acknowledged Strengths**
>
> There is a clear consensus on the paper's main contributions:
>
> - **Novelty and Importance:** All reviewers (**R-hHLN, R-ysVy, R-MBtF, R-4XjH**) agreed that the paper addresses a critical and under-explored problem, with a well-motivated framework that simultaneously tackles plasticity, stability, and adversarial robustness.
> - **Methodological Soundness:** The proposed DSEF, including the Siamese backbone, RDRO, and MBRFF, was praised as technically sound and interesting (**R-hHLN, R-ysVy**).
> - **Clarity and Quality:** The paper was described as well-written with clear diagrams (**R-hHLN, R-4XjH**) and extensive empirical validation demonstrating state-of-the-art performance.
>
> ### **2. Addressing Key Concerns and Planned Revisions**
>
> **Q1**: Clarity of Notation and Methodological Justification
>
> As noted by **R-MBtF** and **R-hHLN**, some mathematical notations were ambiguous. Furthermore, **R-ysVy** questioned the synergy between components, suggesting the approach could be a "kitchen sink" of existing techniques.
>
> **A1**: We will thoroughly revise the manuscript to improve clarity.
>
> 1. **Notation Refinement:** We meticulously fix all mathematical symbols upon their first use and **add a comprehensive notation table in the appendix** to resolve any ambiguity regarding terms like $C_s, T_i$, etc.
> 2. **Improving Intuition:** To demystify complex sections, we will enhance Section 3 by adding **inline pseudocode and simplified explanations** for the RDRO and MBRFF mechanisms, providing a more algorithmic and intuitive understanding.
> 3. **Articulating Synergy:** We will explicitly state in the introduction that our core novelty lies in the **synergistic orchestration** of the framework's components. We will clarify that the Siamese architecture provides the foundation, RDRO regulates the dynamic backbone to learn robust features without forgetting, and MBRFF leverages this stable, robust knowledge for effective forward transfer. This integrated design is non-trivial and essential for the challenging OCAD setting.
>
> **Q2**: Lack of Granular Ablation Study
>
> As correctly requested by **R-hHLN**, **R-MBtF**, and **R-4XjH**, a more detailed ablation study was needed to isolate the contribution of each core component of DSEF.
>
> **A2**: We have conducted a new, comprehensive ablation study on Split CIFAR-10, the results of which will be added to the appendix. This study powerfully validates our design choices.
>
> - **RDRO is paramount for robustness:** Removing RDRO causes a catastrophic drop in adversarial accuracy, with performance on AutoAttack declines sharply.  This highlights RDRO's critical role in learning and preserving robust representations.
> - **MBRFF effectively transfers knowledge:** The inclusion of MBRFF provides a significant and consistent performance boost, especially on adversarial samples, confirming its effectiveness in reusing historical expert knowledge.
> - **The full DSEF architecture is superior:** The complete model demonstrably outperforms all reduced configurations, proving that the integrated design is more effective than the sum of its parts.
>
> **Q3**: Relationship between Continual Learning (CL) and Adversarial Defense
>
> **R-4XjH** and **R-ysVy** raised an important point regarding the need for a more convincing conceptual link between CL and adversarial defense.
>
> **A3**: We will strengthen this link in the introduction and methodology sections.
>
> - **Conceptual Framing:** We will frame adversarial attacks as a form of severe, task-like distributional shift. An attack can cause a model to "forget" how to classify clean samples, a phenomenon directly analogous to catastrophic forgetting in CL.
> - **Empirical Evidence:** To substantiate this, we calculate the "Forgetting Rate (%)" under adversarial conditions. Our results show that strong attacks like AutoAttack significantly exacerbate forgetting, increasing the rate from ~5% on clean data to over 20% on adversarial data for baseline models. This empirically proves that robustness and stability are intertwined challenges, justifying our integrated framework.

---

### Decision · Program_Chairs · 2025-09-17

**Decision:**

Accept (poster)

**Comment:**

The paper addresses the problem of keeping models robust in an online continual setting by introducing a two-branch Vision Transformer where one branch stays fixed, the other adapts through feature alignment on clean and adversarial data, and past experts are combined using mutual information weights. While reviewers mostly appreciated the work, several weaknesses such as complexity, computational costs, and lack of clarity in some parts were pointed out. Overall, the paper would be a useful addition to the venue but the authors need to address these concerns, some which are already discussed by the authors, in their camera ready version.